# Olfaction in fruit flies *(Tephritidae)* balances detection and discrimination of host fruits
Gaëlle Ramiaranjatovo [1,2,3], Maud Charlery de la Masselière[1], Teun Dekker[3], Pierre-François Duyck [4,5], Sebastian Larsson Herrera [6], Bernard Reynaud [1,2] & Vincent Jacob [1] ✉

Phytophagous insects locate suitable hosts through volatile compounds. Polyphagous species face a particular challenge because their hosts emit diverse chemical profiles, yet their olfactory strategies remain unclear. A long-standing assumption suggests that these insects respond primarily to compounds shared across hosts. Here we show that olfactory responses of various polyphagous fruit fly species (Tephritidae) are instead tuned to species-specific fruit compounds from 28 host fruits. This tuning translates into a behavioural preference for species-specific over shared fruit compounds, but only at low doses. Previously, response probability in the same species had been reported to be tuned to shared fruit compounds. To reconcile these observations, we propose a working hypothesis, supported by a computational model: an inverse relationship between olfactory response amplitude and probability may have evolved under the ecological need to detect and discriminate hosts. Together, these results highlight how polyphagous Tephritidae balance detection and discrimination through finely tuned olfactory mechanisms. This insight not only advances our understanding of host selection in polyphagous insects but also has potential applications for ecological management and pest control strategies.

Phytophagous insects need to find a suitable host plant to fulfil their primary functions such as feeding, mating and oviposition. They rely on olfaction to detect and identify host by their volatile compounds[1,2]. Their specialised peripheral olfactory system enables them to detect these compounds and integrate the signals from their environment. However in a natural setting there is a miasma of volatile compounds, which means that identifying a host is a complex task requiring an effective strategy. Host cues can consist of specific compounds or of characteristic blends and ratios[3,4]. Understanding these olfactory strategies involves deciphering both the chemical and sensory components of the communication system, which may facilitate the development of novel odour-based pest control strategies[5].

Polyphagous species which exploit a wide host range face a complex olfactory challenge. A widely accepted theory suggests that they rely primarily on volatile compounds shared among their hosts as these could serve as 'chemical bridges'. A few studies support this view showing that some phytophagous *Miridae* or *Tephritidae* are attracted to compounds emitted by multiple host species[6–9]. Yet it does not account for the ability of polyphagous species to discriminate among their hosts, nor for the fact that

species with similar host ranges often exhibit different host preferences[10–12]. Furthermore host choice in these species can shift depending on experience[13–15]. We therefore hypothesise that in addition to shared compounds facilitating host detection, host discrimination relies on species-specific compounds. How these two selective pressures have shaped olfactory detection remains unresolved.

The family *Tephritidae* which includes several major agricultural pests, provides a relevant model for addressing this issue, as it includes a substantial contingent of polyphagous species[16–18]. Some have an extremely wide host range such as the Oriental fruit fly *Bactrocera dorsalis*, which has been reported from over 600 host plants[19–21]. As expected for highly polyphagous species host selection plasticity has also been reported[13,14,22,23]. Previous work by Biasazin and colleagues demonstrated that some polyphagous fruit fly species from this family detect preferentially and are strongly attracted to compounds that are shared between different host fruits[8].

Here we further investigate how the peripheral olfactory system of polyphagous *Tephritidae* has adapted to the dual ecological challenges of

[1]UMR PVBMT, CIRAD, La Réunion, France. [2]UMR PVBMT, Université de La Réunion, La Réunion, France. [3]Chemical Ecology Unit, Department of Plant Protection Biology, Swedish University of Agricultural Sciences, Alnarp, Sweden. [4]UMR PVBMT, CIRAD, Nouméa, New Caledonia. [5]Equipe ARBOREAL, IAC, Nouméa, New Caledonia. [6]Hushållningssällskapet Skåne, Kristianstad, Sweden. ✉e-mail: vincent.jacob@cirad.fr

host detection and host discrimination. First we analysed the headspaces of 28 highly divergent species of host fruit using gas chromatography coupled with mass spectrometry (GC–MS) and quantified compound's degree of sharedness among their volatile emissions. Since fruit flies lay eggs not only in intact fruit but also on pre-existing mechanical wounds we sampled both undamaged fruit still on the tree and sliced fruits, as the latter may release additional attractive volatiles. Then we mapped the antennal olfactory responses of females from eight *Tephritidae* species—*B. dorsalis, Bactrocera zonata, Dacus demmerezi, Ceratitis capitata, Ceratitis catoirii, Ceratitis quilicii, Neoceratitis cyanescens* and *Zeugodacus cucurbitae*[19,24]—to a set of synthetic fruit compounds, selected along the range of the degree of sharedness. Olfactory responses to odorant stimulations delivered both as direct odour puffs or through chopper-modulated GC were tested using triple electroantennography (EAG$_3$, or EAD$_3$ for EAG$_3$ detector)[25].

Then we developed a computational model of the *Tephritidae* peripheral olfactory system to explore which olfactory tuning properties support efficient host detection and discrimination and might have been selected. By integrating fruit headspace data to capture the natural distribution of compounds, the model allowed us to test how the probability and amplitude of antennal responses to species-specific and shared fruit compounds affect host detection and discrimination.

Our results revealed that antennal response amplitudes are tuned to species-specific compounds rather than to shared ones, consistent with the model's predicted strategies for efficient discrimination. Moreover our modelling suggests that two tuning properties can coexist under selection pressures favouring both host detection and discrimination: enhanced sensitivity to species-specific compounds, as shown here and detection of a larger proportion of shared compounds, as previously reported[8].

Lastly these findings prompted us to design a behavioural assay with simplified, artificial stimuli, in order to probe whether the peripheral sensitivity bias could manifest at the behavioural level. Rather than mimicking natural ecological conditions we tested the behavioural preference of female *B. dorsalis* using a dual-choice trap ex situ, offering two sets of volatiles with a contrasting degree of fruit sharedness. We observed a dose-dependent shift in behavioural preference between species-specific and shared fruit compounds consistent with the hypothesis of a greater olfactory sensitivity to species-specific compounds.

By combining chemical analysis electrophysiology, computational modelling and behavioural assays, we provide a mechanistic framework to understand how polyphagous *Tephritidae* olfactory systems are tuned to balance host detection and discrimination.

## Results
### The volatile emissions of many fruit species partially overlap
To explore the diversity of the volatile emissions of 28 fruit species using ATD-GC–MS we analysed both intact fruit (still on the tree) and sliced fruit (Fig. 1). For intact fruit we detected 511 compounds after excluding 130 found in negative controls. Among them 379 were tentatively identified and a further 55 were unidentified, but assigned to a chemical class. For the sliced fruit we detected 665 compounds after excluding 151 found in negative controls. Among them 463 were tentatively identified and a further 38 were assigned to a chemical class. The main chemical classes were esters (13.31% and 18.35% of the total number of compounds for intact and sliced fruit, respectively) and terpenoids (14.87% and 13.38% of the total number of compounds for intact and sliced fruit, respectively). The remaining compounds were classified as: alcohol, aldehyde, amine, aromatic, carboxylic acid, ether, furanoid, green leaf volatile (GLV), hydrocarbon, ketone, organohalogen, organophosphorus, organosulfur and unknown (Fig. 1E). The proportions of the different chemical families of compounds in intact and sliced fruit were significantly different ($\chi^2$ (15) = 31.084; $p$ = 0.008). Supplementary Data 1 and 2 provide the list of all the identified compounds.

Since samples were stored at room temperature before analysis some degree of chemical degradation may have occurred. However this had minimal impact on the overall characterisation of emitted compounds

(Supplementary Fig. 1). Multivariate analysis confirmed that variability within fruit species was low compared to variability between species (permanova, for intact fruits samples, $R^2$ = 0.63, $F$ = 10.5, $p$ = 0.001, for sliced fruits samples, $R^2$ = 0.74, $F$ = 15.4, $p$ = 0.001).

Despite the great diversity of fruit species included in this study fruit emissions overlapped partially, with compounds emitted by 1–27 fruit species. For each compound we calculated indices to quantify the degree of sharedness among the volatile emissions of intact (sharedness$_{IF}$[28]) and sliced (sharedness$_{SF}$[28]) fruit. Both indices differed significantly between chemical classes (Kruskall–Wallis test, for intact fruit, $\chi^2$ (15) = 45, $p < 10^{-4}$, for sliced fruits, $\chi^2$ (14) = 63, $p < 10^{-7}$). For instance terpenoids and GLVs tended to have higher degree of sharedness among fruits than other compounds (Fig. 2A, B). If there is a strong phylogenetic signal closely related fruit species are likely to emit the same compounds. To test if compounds with a high degree of sharedness result from this type of phylogenetic auto-correlation, we computed the phylogenetic signal using the Abouheif's $C_{mean}$ index for intact ($C_{mean, IF}$[28]) and sliced ($C_{mean, SF}$[28]) fruit samples. Indeed we found a significant correlation between sharedness$_{IF}$[28] and $C_{mean, IF}$[28] (linear regression, $F$ (1, 509) = 13.6; $r$ = 0.16 [c.i. 0.07 –0.24]; $p$ (bootstrap) < 0.001) and between sharedness$_{SF}$[28] and $C_{mean, SF}$[28] (linear regression; $F$ (1, 663) = 19.3; $r$ = 0.17 [c.i. 0.1–0.24]; $p$ (boostrap) < 0.001, Fig. 2C, D). This was mostly led by a significant correlation for terpenoids (linear regressions, intact fruit emissions, $F$ (1,91) = 11.5, $r$ = 0.34 [c.i. 0.16–0.50], $p$ (bootstrap) < 0.001; sliced fruit emissions, $F$ (1, 98) = 12, $r$ = 0.32 [c.i. 0.12–0.51], $p$ (bootstrap) < 0.001), ethers (linear regressions, intact fruit emissions, $F$ (1, 11) = 1.32, $r$ = 0.33 [c.i. 0.06–0.81], $p$ (bootstrap) = 0.022; sliced fruit emissions, $F$ (1, 17) = 2, $r$ = 0.32 [c.i. 0.05–0.67], $p$ (bootstrap) = 0.02) and hydrocarbons in sliced fruit emissions (linear regressions, $F$ (1, 42) = 3.6, $r$ = 0.28 [c.i. 0.08–0.48], $p$ (bootstrap) = 0.012), while the correlation was not significant for the other chemical classes.

Since the atmospheric lifetime (AL) of volatile compounds which depends on their chemical nature, has been shown to correlate with their ecological significance[26], we also investigated whether the degree of sharedness among fruit correlates with AL. Both sharedness$_{IF}$[28] and sharedness$_{SF}$[28] were significantly correlated with AL when accounting for the effect of chemical class as shown by partial correlation analysis (ANOVA, for sharedness$_{IF}$[28], $F$ (1, 294) = 16.6, $\rho$ = 0.23 [c.i. 0.13– 0.34], $p$ (bootstrap) < 0.001; for sharedness$_{SF}$[28], $F$ (1, 353) = 5.4, $\rho$ = 0.12 [c.i. 0.004–0.25], $p$ (bootstrap) = 0.04, Fig. 2E, F). Additionally sharedness$_{SF}$[28] depended significantly on the interaction effect of AL and chemical class (ANOVA, $F$ (13, 353) = 2.7, $p$ = 0.001), indicating that not all classes contributed equally to this relationship. When analysing each class independently sharedness$_{IF}$[28] was significantly correlated with AL for alcohols (linear regression, $F$ (1, 37) = 7.2, $r$ = 0.4 [c.i. 0.1–0.67], $p$ (bootstrap) = 0.008), esters (linear regression, $F$ (1, 65) = 10, $r$ = 0.37 [c.i. 0.09–0.58], $p$ (bootstrap) = 0.012) and ketones (linear regression, $F$ (1, 37) = 7.2, $r$ = 0.68 [c.i. 0.27–0.91], $p$ (bootstrap) = 0.006). Similarly sharedness$_{SF}$[28] was significantly correlated with AL for aldehydes (linear regression, $F$ (1, 28) = 6.4, $r$ = 0.43 [c.i. 0.04–0.71], $p$ (bootstrap) = 0.034) and ketones (linear regression, $F$ (1, 24) = 11.5, $r$ = 0.58 [c.i. 0.16–0.88], $p$ (bootstrap) = 0.01).

### *Tephritidae* olfactory responses are negatively correlated with the degree of sharedness of volatile compounds among fruit species
A set of 38 commercially available compounds chosen to be regularly distributed along the fruit sharedness indices and whose identities were confirmed by co-injection with standards, was used to test the olfactory system of *Tephritidae* species (Supplementary Table). We chose compounds from two chemical classes only esters and terpenoids, given that we intended to explore the role of ecological rather than chemical variability. For these compounds the indices sharedness$_{IF}$[28] and sharedness$_{SF}$[28] were significantly correlated ($F$ (1, 32) = 26.7; $p < 10^{-4}$). Moreover sharedness$_{SF}$[28] was not correlated with the compounds main chemical properties, namely boiling point ($F$ (1, 36) = 3.3; $p$ = 0.08), vapour pressure ($F$ (1, 36) = 1.9; $p$ = 0.18), lipophilicity estimated with log *Poct/wat* (ChemSketch, ACD/Labs) ($F$ (1,

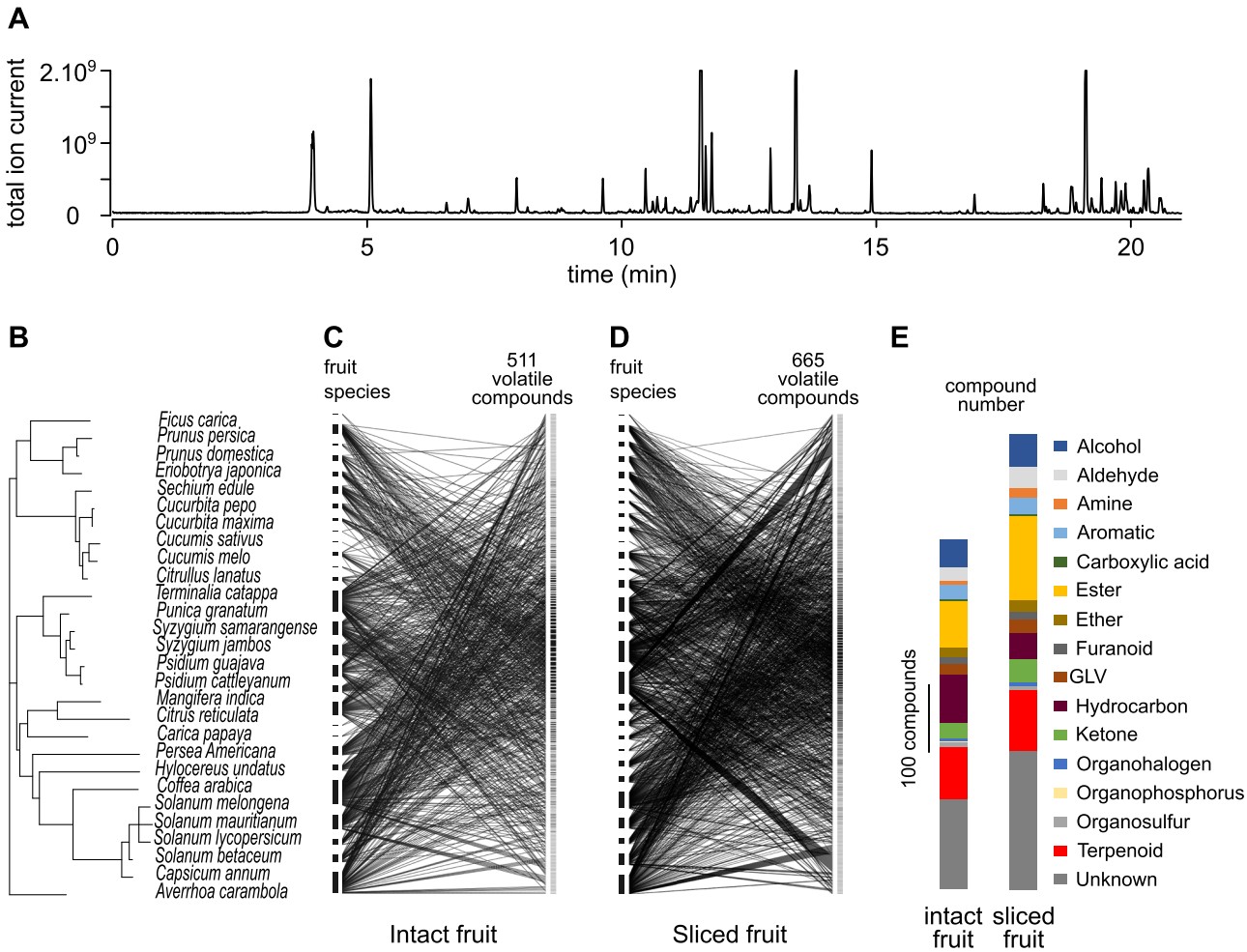

**Fig. 1 | Diversity of volatile emissions produced by the 28 host fruit species of the** *Tephritidae* **studied. A** Example of a chromatogram collected from the volatile emissions of a guava fruit (*Psidium guajava*). **B** Phylogeny of the 28 fruit species from which volatile emissions were collected. **C** Interaction network between the fruit species (left) and the 511 volatile compounds (right) collected from intact fruit. The fruit species are sorted in the same phylogenetic order as in (**A**) and the compounds are sorted by the degree of shardedness among fruits, for which the maximal number is in the middle and the minimal (1 fruit) is at both extremities. **D** Interaction network between the fruit species (left) and the 665 volatile compounds (right) collected from sliced fruit, ordered as in (**C**). **E** Chemical classification of volatile compounds tentatively identified for the 28 fruit species. The two bars show the compounds of intact fruit (left) and sliced fruit (right), respectively. Each colour represents the chemical classes of compounds and the corresponding height is proportional to the number of compounds in each chemical class.

36) = 0.62; $p$ = 0.44), or depression rate[27] ($F_{(1, 36)}$ = 1.3; $p$ = 0.26) (Supplementary Fig. 2).

The antennal responses to 30 of the compounds were measured in females from eight *Tephritidae* species with EAG$_3$ at a dose of $10^{-4}$ (dilution v/v). Antennal response to negative control were recorded initially (mean 7.2 CSD units, bootstrapped 95% c.i.: 6.2–8.5), after the 10th stimulation with a compound (mean 5.9, bootstrapped 95% c.i.: 4.8–7.2), after the 20th stimulation with a compound (mean 6.6, bootstrapped 95% c.i.: 5.5–7.8) and at the end of the experiment (mean 6.9, bootstrapped 95% c.i.: 5.7–8.0), with values presented for all species combined. As the responses did not increase, contamination was unlikely. Also, antennal response to 1-octen-3-ol was tested initially as a positive control (mean response after subtracting response to negative control 11.8, bootstrapped 95% c.i.: 9.6–14.2). We then tested how EAG$_3$ response correlated with shardedness$_{SF}$[28], using each individual and the compound depression rate, AL and chemical class as independent variables to rule out possible experimental bias (partial correlation analysis). Antennal response was negatively correlated with shardedness$_{SF}$[28] (linear regression, $F_{(1, 975)}$ = 38, $\rho$ = −0.19 [c.i. −0.25 to −0.13], $p$ (bootstrap) < 0.001), with an interaction with species identity ($F_{(7, 975)}$ = 2; $p$ (ANOVA) = 0.05). More precisely it was significant in all polyphagous species tested independently: *B. dorsalis*, $n$ = 4 individuals,

linear regression, $F_{(1, 112)}$ = 7.6, $\rho$ = −0.25 [c.i. −0.42 to −0.05], $p$ (bootstrap) = 0.018; *B. zonata*, $n$ = 4 individuals, $F_{(1, 112)}$ = 10.1, $\rho$ = -0.29 [c.i. −0.44 to −0.1], $p$ (bootstrap) = 0.002; *C. capitata*, $n$ = 5, $F_{(1, 141)}$ = 19.7, $\rho$ = −0.35 [c.i. −0.48 to −0.22], $p$ (bootstrap) < 0.001; *C. catoirii*, $n$ = 5, $F_{(1, 141)}$ = 5, $\rho$ = −0.19 [c.i. −0.34 to −0.01], $p$ (bootstrap) = 0.032; *C. quilicii*, $n$ = 3, $F_{(1, 83)}$ = 5.4, $\rho$ = −0.25 [c.i. −0.43 to −0.05], $p$ (bootstrap) = 0.014 (Fig. 3). However it was not significant in the three oligophagous species: *D. demmerezi*, $n$ = 4, $F_{(1, 112)}$ = 1.8, $\rho$ = −0.13 [c.i. −0.33–0.08], $p$ (bootstrap) = 0.25; *N. cyanescens*, $n$ = 5, $F_{(1, 141)}$ = 2.85, $\rho$ = −0.14 [c.i. −0.29 to 0.02], $p$ (bootstrap) = 0.1; and *Z. cucurbitae*, $n$ = 4, $F_{(1, 112)}$ = 2.63, $\rho$ = −0.15 [c.i. −0.31 to 0.01], $p$ (bootstrap) = 0.07. Antennal response was also negatively correlated with shardedness$_{IF}$[28] ($F_{(1, 839)}$ = 6.8, $\rho$ = −0.09 [c.i. −0.16 to −0.02], $p$ (bootstrap) = 0.008), with an interaction with species identity ($F_{(7, 839)}$ = 2.5; $p$ (ANOVA) = 0.01). *B. dorsalis* was the only species assessed at two doses (10 µL, $10^{-4}$ and $10^{-2}$ v/v; $n$ = 4 and 7 individuals, respectively). When the two doses were combined antennal responses remained negatively correlated with shardedness$_{SF}$[28] ($F_{(1, 314)}$ = 7.9, $\rho$ = −0.19 [c.i. −0.29 to −0.07], $p$ (bootstrap) = 0.002, $p$ = 0.005) and with shardedness$_{IF}$[28] ($F_{(1, 270)}$ = 4, $\rho$ = −0.1 [c.i. −0.21 to 0.03], $p$ (bootstrap) = 0.13, $p$ = 0.05), with no significant interaction with dose ($F_{(1314)}$ = 0.32, $p$ = 0.57 and $F_{(1270)}$ = 1.1, $p$ = 0.29, respectively, Supplementary Fig. 3).

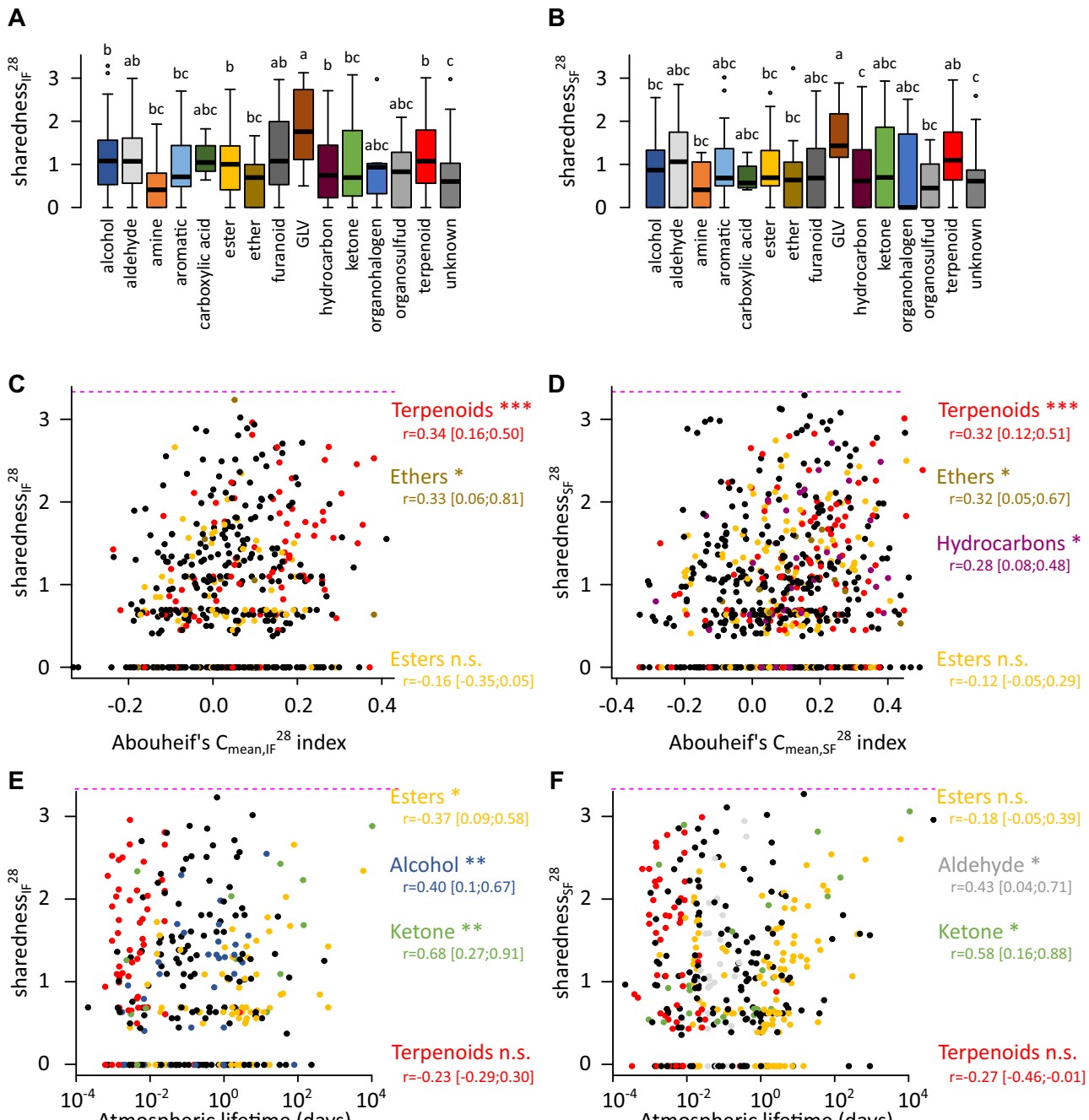

**Fig. 2 | Relationship between fruit sharedness indices, chemical class, fruit phylogenetic signal and atmospheric lifetime. A**, **B** The boxplots show the distribution of sharedness indices for intact (**A**) and sliced (**B**) fruit emissions for each chemical classes. The number of compounds within each class in shown in Fig. 1E. Two boxes with the same letter are not significantly different (Dunn test with Benjamini–Hochberg correction for multiple comparisons, $\alpha = 0.05$). **C**–**F** Relationship between fruit sharedness and phylogenetic signal Aubouheif's $C_{mean}$, or atmospheric lifetime. Each point represents a compound for the emissions from intact (**C**, **E**) and sliced (**D**, **F**) fruit. Compounds with a sharedness index of 0 are emitted by only one fruit species. The magenta dashed line indicates the theoretical maximal value reached by the sharedness indices if all fruit species emit the compound. A $C_{mean}$ of 0 means phylogenetic independence, while values above 0.2 are considered phylogenetically correlated. For readability, the colour code was limited to the two main chemical classes, terpenoids and esters, as well as to the chemical classes showing a significant correlation. All other classes are shown in black. Pearson's correlation coefficients ($r$) are provided along with their bootstrapped 95% confidence intervals (in brackets). Boostrap test for positive correlation: *, $p < 0.05$, **, $p < 0.01$, ***, $p < 0.001$, n.s., not significant.

The number of fruit species considered was of little consequence: we calculated indices of sharedness among the volatile emissions of a subset of 13 fruit species (Table 1), reported to have a *B. dorsalis* infestation rate of above 5% in La Réunion (named sharedness$_{IF}^{13}$ and sharedness$_{SF}^{13}$ for intact and sliced fruit, respectively). We found a significant correlation between sharedness$_{IF}^{13}$ and sharedness$_{IF}^{28}$ ($F_{(1, 32)} = 87$; $r = 0.86$ [c.i. 0.71–0.95], $p$ (bootstrap) < 0.001) and between sharedness$_{SF}^{13}$ and sharedness$_{SF}^{28}$ ($F_{(1,}$ $36) = 122.7$; $r = 0.88$ [c.i. 0.74–0.97], $p$ (bootstrap) < 0.001, Supplementary Fig. 4). Again antennal response was negatively correlated with sharedness$_{SF}^{13}$ ($F_{(1, 975)} = 47$, $\rho = -0.21$ [c.i. −0.27 to −0.15], $p$ (bootstrap) < 0.001), without a clear interaction with species identity ($F_{(7,} $ $975) = 1.9$; $p = 0.06$). It was significant in most species tested individually: *B. dorsalis*, $F_{(1, 112)} = 2.63$, $\rho = -0.28$ [c.i. −0.44 to −0.09], $p$ (bootstrap) = 0.01; *B. zonata*, $F_{(1, 112)} = 15.8$, $\rho = -0.35$ [c.i. −0.52 to −0.15],

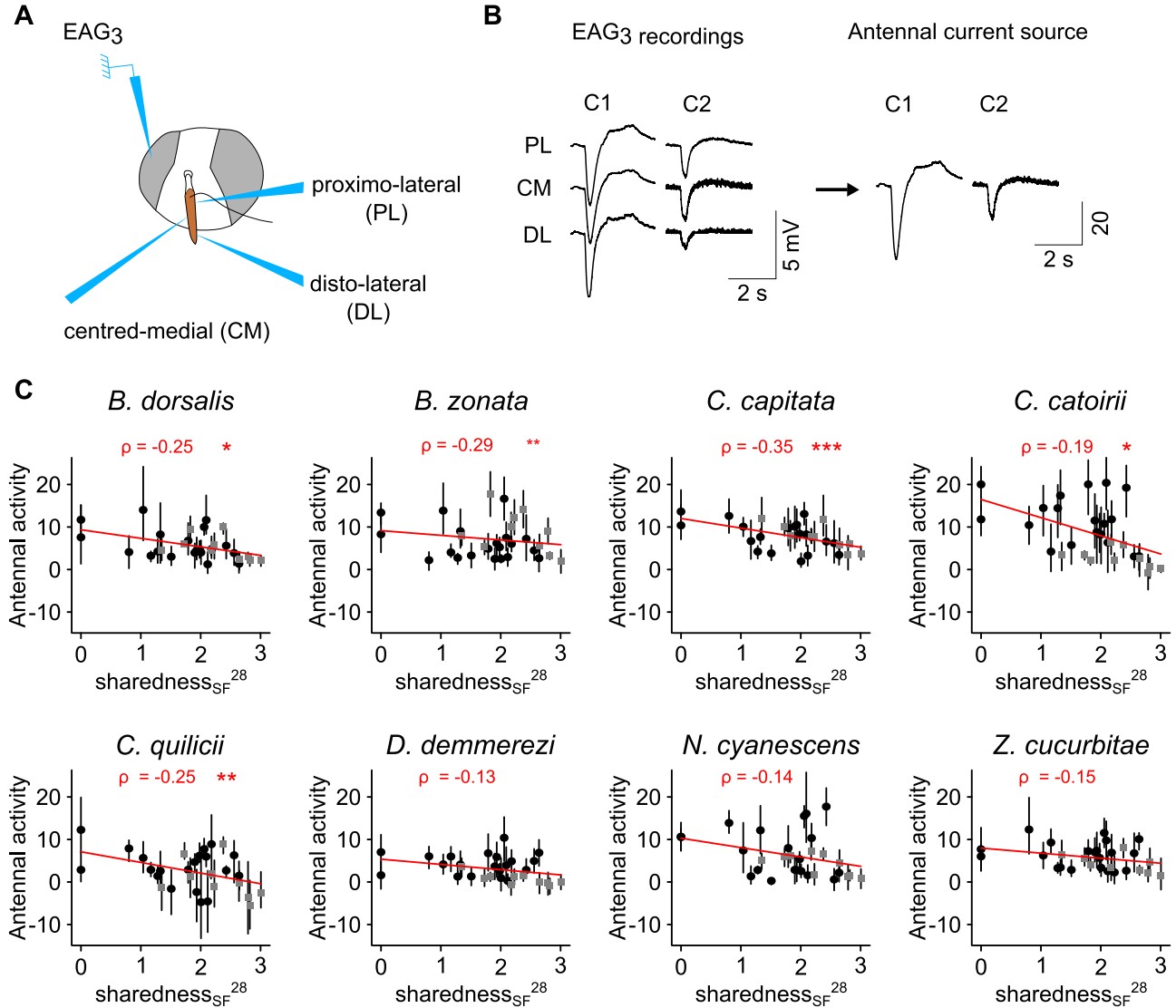

**Fig. 3 | Antennal responses of eight *Tephritidae* species correlates with the degree of sharedness among fruits. A** Schematic representation of the EAG$_3$ mounting. **B** An example of EAG$_3$ recording on individual female *B. dorsalis* stimulated with two compounds (C1 and C2). EAG signals are simultaneously recorded at three positions on the antenna and used to estimate the total antennal activity through a corresponding antennal current source inferred from a CSD model. **C** Antennal responses of females from eight *Tephritidae* species to synthetic compounds at

dilution v/v: $10^{-4}$ depends on the degree of sharedness among sliced fruits. Circles (esters) and grey squares (terpenoids): mean value; bars: bootstrapped 95% c.i., $n = 3-5$ individuals per species. Red lines correspond to linear regression curves whose partial Pearson's correlation coefficient ($\rho$, with correction for depletion rate, AL and chemical class) and significance is shown (*$p < 0.05$; **$p < 0.01$; ***$p < 0.001$, bootstrap bilateral test).

$p$ (bootstrap) $< 0.001$; *C. capitata*, $F (1, 141) = 16.6, \rho = -0.32$ [c.i. $-0.45$ to $-0.20$], $p$ (bootstrap) $< 0.001$; *C. quilicii*, $F (1, 83) = 13.4, \rho = -0.31$ [c.i. $-0.46$ to $-0.13$], $p$ (bootstrap) $= 0.002$; *D. demmerezi*, $F (1, 112) = 4.9$, $\rho = -0.27$ [c.i. $-0.44$ to $-0.10$], $p$ (bootstrap) $= 0.022$; and *Z. cucurbitae*, $n = 4, F (1, 112) = 4.9, \rho = -0.20$ [c.i. $-0.37$ to $-0.02$], $p$ (bootstrap) $= 0.004$. It was however not significant in *C. catoirii*, $n = 5, F (1, 141) = 0.7, \rho = -0.07$ [c.i. $-0.23$ to $0.11$], $p$ (bootstrap) $= 0.47$ and *N. cyanescens*, $F (1, 141) = 8.8$, $\rho = -0.16$ [c.i. $-0.33$ to $0.04$], $p$ (bootstrap) $= 0.104$. Antennal response was also negatively correlated with sharedness$_{IF}$[13] ($F (1, 839) = 8.1, \rho = -0.10$ [c.i. $-0.17$ to $-0.03$], $p$ (bootstrap) $= 0.002$), with an interaction with species identity ($F (7, 839) = 2.3; p = 0.03$).

We then explored the EAD$_3$ antennal response to 35 compounds. EAG response to negative and positive control stimulations applied before and after the GC run were similar in amplitude, confirming the stability of preparation responsiveness. Across all species the mean antennal response to the negative control was 8.4 CSD units [c.i. 7.4–9.5] before and 8.5 [c.i. 7.5–9.6] after the run. For the positive controls after subtracting the negative

control response, the mean antennal response to 1-octen-3-ol was 31.4 [c.i. 24.7–38.2] before and 28.1 [c.i. 22.2–34.6] after the run and the mean antennal response to hexenyl acetate was 26.2 [c.i. 20.3–32.2] before and 23.2 [c.i. 18.2–29.0] after the run. For the other compounds the stimulations were provided through chopping modulated GC. This protocol induced more experimental bias since antennal responses to GC-stimulations were strongly correlated with compound depletion rate ($F (1, 264) = 23, \rho = 0.28$ [c.i. 0.11–0.43], $p$ (bootstrap) $< 0.001$) and AL ($F (1, 264) = 40, \rho = 0.36$ [c.i. 0.25–0.47], $p$ (bootstrap) $< 0.001$), which might result from the chopping modulation. However we found that the GC-EAD$_3$ antennal response of the eight species was negatively correlated with sharedness$_{SF}$[28] ($F (1, 263) = 14.4$, $\rho = -0.23$ [c.i. $-0.34$ to $-0.12$], $p$ (bootstrap) $< 0.001$; $p = 0.0002$) and with sharedness$_{IF}$[28] ($F (1, 231) = 16.7, \rho = -0.26$ [c.i. $-0.38$ to $-0.13$], $p$ (bootstrap) $< 0.001$), with no interaction with species identity ($F (7, 263) = 0.37$; $p = 0.92$ and $F (7, 231) = 0.7; p = 0.70$, respectively). For each species tested independently antennal response depended on a joint effect between sharedness$_{IF}$[28] and compound depletion rate (Supplementary Fig. 5):

*B. dorsalis* $n = 4$ individuals, linear regression, $F(1, 27) = 7.6$; $p = 0.01$; but *B. zonata*, $n = 4$, $F(1, 27) = 3.8$; $p = 0.06$; *C. capitata*, $n = 3$, $F(1, 27) = 11$; $p = 0.003$; *C. catoirii*, $n = 3$, $F(1, 27) = 4.5$; $p = 0.04$; *C. quilicii*, $n = 3$, $F(1, 27) = 8.8$; $p = 0.006$; *D. demmerezi*, $n = 3$, $F(1, 27) = 7.3$; $p = 0.01$; *N. cyanescens*, $n = 3$, $F(1, 27) = 8.5$; $p = 0.007$; and *Z. cucurbitae*, $n = 4$, $F(1, 27) = 22.8$; $p < 10^{-4}$. The effect was similar at a higher dose (1 µg) tested in *B. dorsalis*: antennal response depended on a joint effect between sharedness$_{IF}$[28] and compound depletion rate ($n = 5$ individuals, linear regression, $F(1, 27) = 6.6$; $p = 0.02$). To sum up with regard to GC-EAD$_3$ data the main effect was a negative correlation between antennal response and sharedness$_{IF}$[13] for compounds with a low depletion rate.

## A computational model predicts a contrasted use of species-specific and shared fruit compounds for odour-based fruit detection and discrimination

The amplitude of EAG responses as observed in this study and the probability that a compound induces an EAG response, as reported previously[8], appear to be inversely correlated with compound sharedness among fruit. We hypothesised that the bidirectional property of olfactory tuning in these species was selected both by the ecological need to detect a large number of hosts, encouraging responses to host-shared compounds and the ecological need to discriminate host species, encouraging responses to species-specific compounds. Computational modelling was used to test this. We built 120,000 random neuronal models of a peripheral olfactory system (Fig. 4A). The GC–MS data for intact or sliced fruit samples were used as input odours for these models in order to consider the natural variability of fruit emissions within and between fruit species and the natural distribution of the degree of sharedness of volatile compounds in fruit emissions. For each model we estimated the efficiency of the olfactory system to detect and discriminate fruit using specially designed indices (Fig. 4B). The performance of the models was heterogeneous, allowing us to explore the typical chemical tuning of the most efficient olfactory systems for either detection, discrimination or both. Not surprisingly models with 20 ORs discriminated fruit better than models with 10 ORs ($F(1, \text{df} > 10^5) = 258,197$; $p < 10^{-15}$) and the fruit detectability index was correlated with the number of compounds detected per OR ($F(2, \text{df} > 10^5) = 921,033$; $p < 10^{-15}$). For cross-comparison, we standardised the distributions (expressed as z-scores) for each combination of number of ORs, number of compounds detected per OR and input data sets (intact or sliced fruit emissions).

We calculated for each model the correlation coefficient between a compound's probability of being detected and its degree of sharedness among fruits. Additionally we computed the correlation coefficient between the amplitude of the EAG response to those detected compounds, at four different doses and their degree of sharedness among fruits. By design, the two variables were independent (linear regression, $F(1, \text{df} > 10^5) = 0.02$; $p = 0.89$). The fruit detectability index depended significantly on the two correlation coefficients (linear regression, Fig. 4C, $F(1, \text{df} > 10^5) = 1238$ and $3939$; $p < 10^{-15}$ and $10^{-15}$, respectively) with a significant interaction (linear regression, $F(1, \text{df} > 10^5) = 11$; $p < 0.001$). The interaction effect revealed that models with only one of the two correlation coefficients high performed better than models where the two correlation coefficients had intermediate values. The fruit discriminability index also depended significantly on the two correlation coefficients (linear regression, Fig. 4D, $F(1, \text{df} > 10^5) = 1988$ and $4,020$; $p < 10^{-15}$ and $10^{-15}$, respectively) with a significant interaction (linear regression, $F(1, \text{df} > 10^5) = 5.6$; $p = 0.018$). More specifically the two response parameters (probability and amplitude) were positively correlated with the degree of sharedness among fruits in the most efficient models at detecting fruits and negatively correlated with it in the most efficient models at discriminating fruits. This functional trade-off was robust to the model parameters as it was observed for any OR numbers numbers of detected compounds per OR and intact or sliced fruit input data, or stimulation dose tested (Supplementary Fig. 6). The effect on EAG amplitude depended mostly on the sensitivity thresholds to detected compounds (Supplementary Fig. 7) and, to a lesser extent, the dynamic range of detected compounds.

### Table 1 | List of the 28 fruit species included in this study

| Fruit families | Fruit species | Common names |
|---|---|---|
| Anacardiaceae | *Mangifera indica*[a] | Mango |
| Cactaceae | *Hylocereus undatus*[a] | Dragon fruit |
| Caricaceae | *Carica papaya*[a] | Papaya |
| Combretaceae | *Terminalia catappa*[a] | Indian almond |
| Cucurbitaceae | *Sechium edule* | Chayote |
| Cucurbitaceae | *Cucurbita maxima* | Pumpkin |
| Cucurbitaceae | *Cucumis sativus* | Cucumber |
| Cucurbitaceae | *Cucurbita pepo* | Zucchini |
| Cucurbitaceae | *Cucumis melo* | Melon |
| Cucurbitaceae | *Citrullus lanatus* | Watermelon |
| Lauraceae | *Persea americana*[a] | Avocado |
| Lythraceae | *Punica granatum* | Pomegranate |
| Moraceae | *Ficus carica*[a] | Fig |
| Myrtaceae | *Psidium guajava*[a] | Guava |
| Myrtaceae | *Psidium cattleianum*[a] | Strawberry guava |
| Myrtaceae | *Syzygium samarangense*[a] | Java apple |
| Myrtaceae | *Syzygium jambos*[a] | Rose apple |
| Oxalidaceae | *Averrhoa carambola* | Star fruit |
| Rosaceae | *Eriobotrya japonica*[a] | Loquat |
| Rosaceae | *Prunus persica*[a] | Peach |
| Rosaceae | *Prunus domestica*[a] | Plum |
| Rubiaceae | *Coffea arabica* | Coffee |
| Rutaceae | *Citrus reticulata* | Tangor |
| Solanaceae | *Solanum melongena* | Aubergine/eggplant |
| Solanaceae | *Solanum mauritianum* | Bugweed |
| Solanaceae | *Capsicum annuum* | Chilli |
| Solanaceae | *Solanum betaceum* | Tree tomato |
| Solanaceae | *Solanum lycopersicum* | Tomato |

[a]Indicate the 13 species reported to have a *B. dorsalis* infestation rate of over 5%.

We then calculated an index of joint functionality assessing how much the olfactory system is efficient at doing both functions simultaneously. This index depended on a joint effect between both correlation coefficients (linear regression, Fig. 4E, $F(1, \text{df} > 10^5) = 21$; $p < 10^{-5}$). This effect was robust to the model parameters even though it was weaker for EAG responses to low doses of compound, or for those models with a high proportion of detected compounds (Supplementary Fig. 6). Specifically it points to two possibilities for an olfactory system to be efficient at detecting and discriminating fruit at the same time: probability of detecting compounds and response amplitude should be inversely correlated, in a direction or the other, with the degree of sharedness among fruits.

## Female *B. dorsalis* display a dose-dependent switch in behavioural preference between shared and species-specific fruit compounds

Our antennal recordings revealed a response bias towards species-specific compounds despite previous reports that *Tephritidae* females can be attracted to volatiles broadly shared across host species[8]. These observations suggest that attraction to shared compounds alone may require higher concentrations than when species-specific compounds are involved. Based on this hypothesis we designed a behavioural assay using a dual-choice ex situ trap (Fig. 5A) to determine whether flies preferentially respond to species-specific compounds at low doses. We selected ten fruit compounds, five of which were species-specific fruit

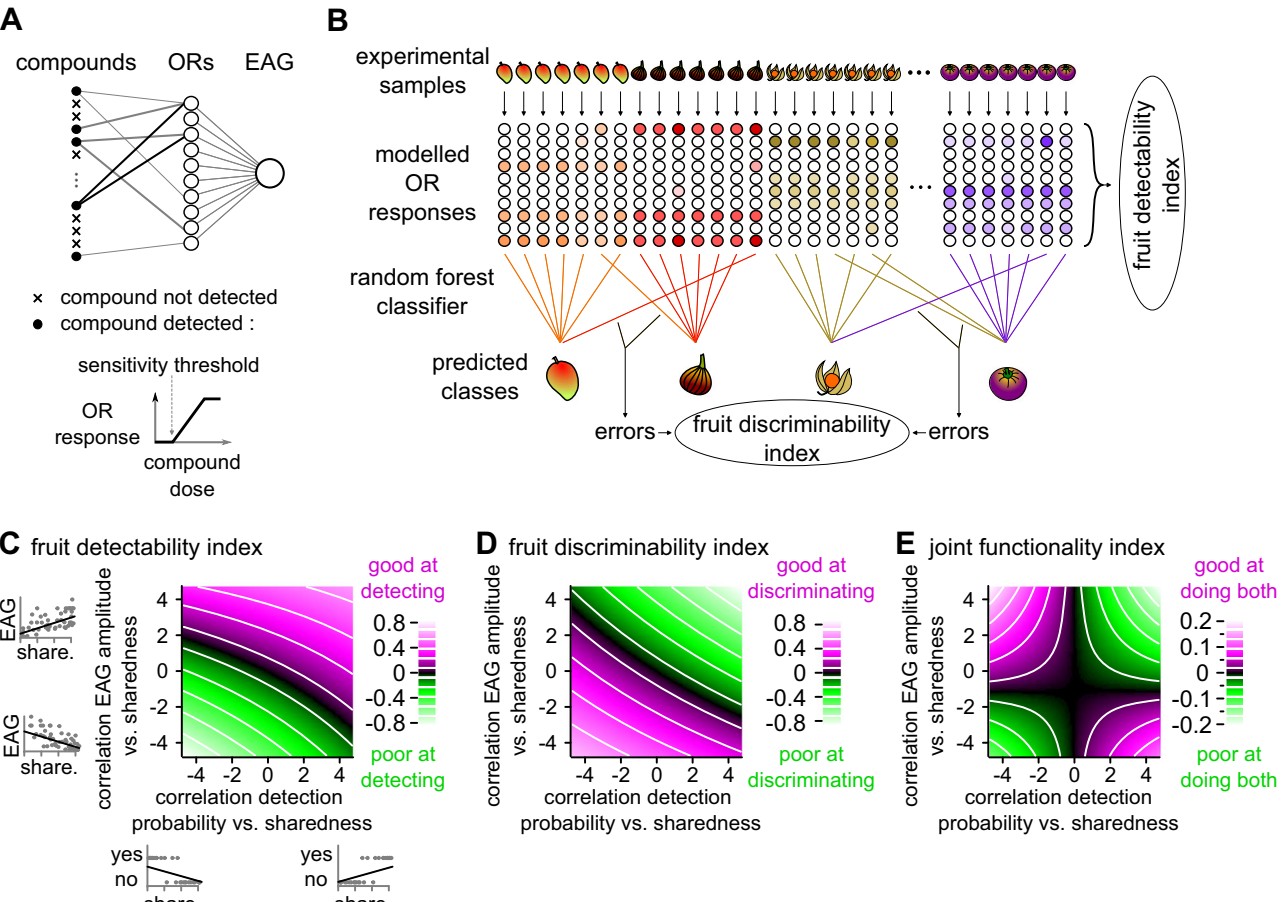

**Fig. 4 | Computational model reveals a functional trade-off between the detection of and discrimination between fruit species. A** Schematic representation of the parameters included in one random model of an insect's olfactory system. Each olfactory receptor (OR) is sensitive to a random set of compounds as shown by a simplified dose–response curve, whose threshold and dynamic range have been drawn randomly. **B** Schematic depiction of the algorithms used to calculate the fruit detectability index and the fruit discriminability index for one random model. For this experimental data corresponding to intact or sliced fruit samples were used as an input (experimental samples) and the modelled OR responses were calculated. The fruit detectability index was the sum of all OR responses to all volatile samples. The fruit-species discriminability index was calculated from the proportion of fruit species misclassified by a random forest algorithm applied on the OR response

patterns. **C** Fruit detectability index (colour code) is correlated both with the correlation coefficient between EAG amplitude and sharedness index (share.) and the correlation coefficient between the probability to detect a compound and sharedness index (linear model $p < 10^{-15}$ in both cases, all variables are expressed in z-score units). **D** Fruit discriminability index (colour code) is correlated both with the correlation coefficient between EAG amplitude and sharedness index and the correlation coefficient between the probability to detect a compound and sharedness index (linear model $p < 10^{-15}$ in both cases, all variables are expressed in z-score units). **E** For models to efficiently detect and discriminate fruit at the same time, amplitude and probability of response should be inversely correlated with the sharedness index (colour code: joint functionality index, all variables are expressed in z-score).

compounds and five shared compounds, all of which were small esters to ensure low chemical variability (Table 2). A preliminary experiment showed that female *B. dorsalis* were more attracted to a blend of the 10 compounds, whose emission rates were homogenised, than to a negative control (GLMM: $\chi^2$ (1, $N = 11$) = 28.845; $p < 0.001$; mean proportion of flies trapped after 125 min: 0.185 (95% c.i.: 0.118–0.252) and 0.003 (95% c.i.: 0.003–0.009), respectively; Fig. 5B). Then we tested the comparative attractiveness of the two sub-blends of five species-specific fruit compounds and five shared compounds. At the highest dose tested (dilution v/v: $10^{-3}$) females preferred the shared compound blend over the species-specific compound blend (GLMM: $\chi^2$ (1, $N = 60$) = 38.355; $p < 0.001$; mean proportion of flies trapped after 240 min: 0.133 (95% c.i.: 0.098–0.168) and 0.109 (95% c.i.: 0.076–0.142), respectively; Fig. 5C). At a lower dose (dilution v/v: $10^{-4}$) a reversal of the trend was observed, i.e. insects were more attracted to the species-specific compound blend than to the shared compound blend (GLMM: $\chi^2$ (1, $N = 46$) = 214.617; $p < 0.001$; mean proportion of flies trapped after 240 min: 0.186 (95% c.i.: 0.118–0.254) and 0.048 (95% c.i.: 0.027–0.070), respectively; Fig. 5D).

## Discussion

In this paper we used chemical analysis and modelling, combined with electrophysiological and behavioural approaches to explore the chemical interactions between *Tephritidae* species and 28 of their host fruit species. By analysing the chemical composition of volatile emissions from 28 *Tephritidae* host fruit either intact (on the tree) or after mechanical damage, we tentatively identified a considerable number of compounds. As expected damaged fruit produced more volatile compounds than intact fruit[28–30]. The volatile emissions were dominated by terpenoids and esters which are characteristic of volatile fruit emissions and are easily detected by insect olfactory receptors[31,32]. Although each fruit species has its own chemical profile there is considerable overlap between species. The degree of sharedness among fruit species varied, some compounds were specific to a single species, while others were emitted by most of the fruit species considered. We found a significant negative correlation between the antennal responses of several *Tephritidae* species and volatile's degree of sharedness among fruits. The effect was robust regardless of how the sharedness index was calculated by considering: all 28 fruit species or a subset of 13 species, with a high infestation rate of generalist species; and intact or sliced fruit.

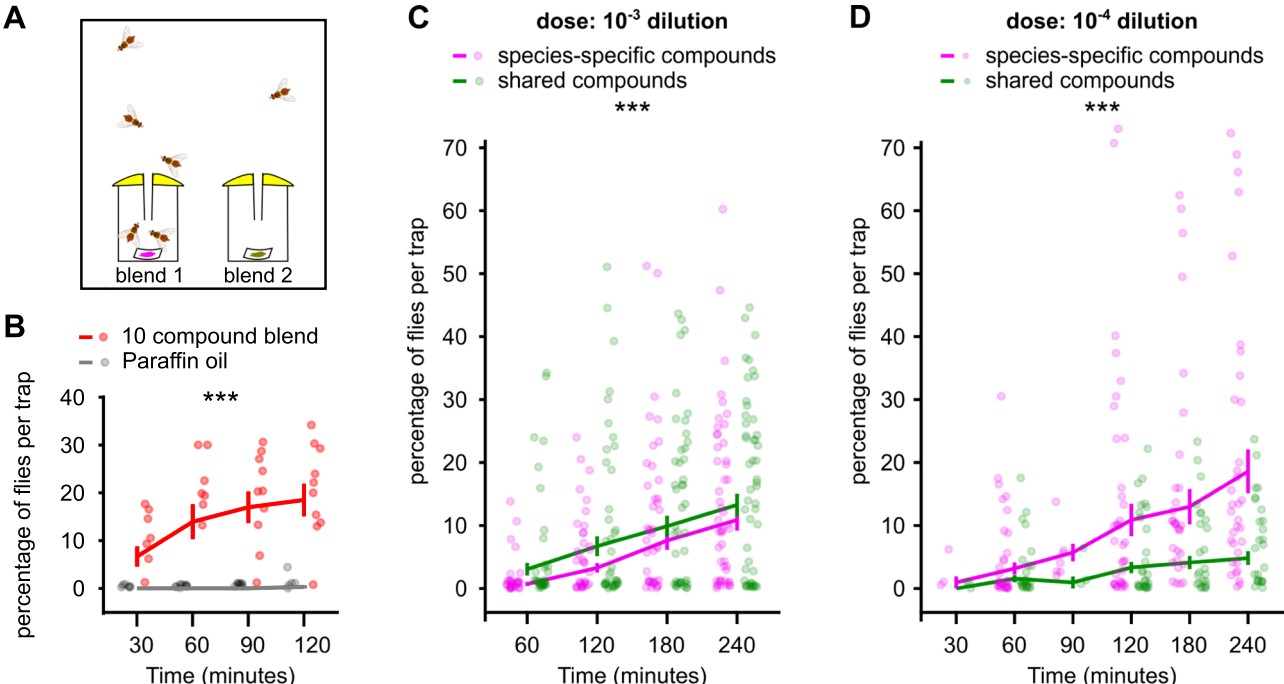

**Fig. 5 | *B. dorsalis* behavioural preference between species-specific and shared fruit compounds is dose-dependent. A** Schematic drawing of the ex situ trapping bioassay. **B** Dual choice tests between a blend of 10 compounds diluted at $10^{-4}$ v/v in mineral oil (red) and mineral oil (black); values are means ± SE ($n = 11$). An artificial time jitter and offset were added to each point for clarity separating the two modalities although they were recorded simultaneously. ***Indicates a significant difference between the two modalities (GLMM, $p < 0.001$). **C** Dual choice tests between a blend of five shared compounds (in green) and a blend of five specific compounds (in magenta) tested at the dilution v/v: $10^{-3}$ ($n = 60$). Same convention as in (**B**). **D** Dual choice tests between the same blends as in panel (**C**) diluted at $10^{-4}$ v/v ($n = 46$). Same conventions as in (**B**).

Lastly we found that mature female *Bactrocera dorsalis* demonstrated a dose-dependent behavioural preference when we compared a blend of species-specific fruit compounds and a blend of shared compounds. This observation is not unexpected: amplitude of EAD responses does not equal the strength of behavioural response since detection is not the same as attraction. Detected volatiles could be deterrents, especially at higher concentrations. By design the attraction observed towards the blend of species-specific compounds cannot be attributed to any shared compounds. Thus the assay confirms that certain species-specific and shared compounds can be behaviourally relevant for the insect, though under different conditions. Further behavioural experiments in more naturalistic contexts will be required to determine how these mechanisms operate under ecological conditions. Below we will discuss how our observations and computational modelling support the alternative hypothesis regarding the olfactory strategy used by *Tephritidae* to explore their wide host range.

The main hypothesis suggests that since *Tephritidae* females have an ecological need to detect a large number of suitable hosts for oviposition, their olfactory system should be tuned for compounds shared by different fruit species. This type of behaviour would allow females to identify a wide host range with minimum effort in terms of information processing and, therefore, minimum energy consumption[6,33,34]. Supporting this theory the addition of three compounds identified in ripe guava considerably increased *Bactrocera tryoni*'s attraction to several non-preferred hosts[35]. The authors suggest that these compounds could be a generic signature for ripe fruit. Indeed the use of particular shared volatiles which indicate ripe fruits is most likely to have been selected, given that many fruit-feeding tephritids exploit ripe fruits essentially. Furthermore a blend of volatile compounds shared between four phylogenetically distant fruit species was attractive for *Bactrocera dorsalis* and *Zeugodacus cucurbitae*[8]. The hypothesis raises a number of issues which we tested. First do phylogenetically distant fruit species emit the same volatile compounds? Clarke predicted a substantial proportion of shared compounds was an adaptation to generalist seed dispersers[17]. We found that the degree of sharedness among fruits was only weakly correlated with fruit phylogeny, i.e.

a number of identical compounds were emitted by fruit that did not belong to the same clade. More specifically the relationship depended on the chemical class: while the degree of sharedness of weakly polar compounds, such as terpenoids, ethers and hydrocarbons, was correlated with fruit phylogeny, the degree of sharedness of small polar compounds, such as esters, alcohol, ketones and aldehydes, was instead linked to their AL, a property that is determined by how easily these compounds undergo oxidative degradation. These observations suggest a complex evolutionary trajectory for fruit emissions with selection pressures that remain to be understood. Anyway relying on shared compounds could explain why tephritid host associations do not correlate with fruit phylogeny[24]. Second do generalist species prefer shared compounds over species-specific fruit compounds? Our observations reveal a preference for shared compounds in *B. dorsalis*, but only for the highest dose tested. Third do shared compounds induce a stronger antennal response and, therefore, are they overrepresented compared to species-specific compounds? Our computational model confirms that this should be the case and suggests two testable predictions: the proportion of compounds detected by the olfactory systems should correlate positively with the compound's degree of sharedness among fruit species, which was indeed observed in another study on *Tephritidae*[8]; the response amplitude to detected compounds should also correlate positively with the compound's degree of sharedness among fruit species, but here we observed a negative correlation. Our observation is unlikely to be an experimental artefact because a bias due to the heterogeneous volatility of the tested compounds should not make a difference. Indeed we found no correlation between sharedness indices and boiling point, lipophilicity or depletion rate[27]. In addition the effect was observed both with a passive and GC-induced vaporisation of compounds. The use of triple electroantennography reduces the risk of uneven sampling of the olfactory receptor[23,24]. Artefactual responses to impurities have been reported[36,37] but our GC-EAD analysis of the same compounds make this explanation unlikely.

When faced with various sources of volatile compounds insects must be able to identify their hosts' volatile compounds from a complex

**Table 2 | List of compounds used for behavioural tests**

| Blend | CAS | Compound | Found in the emissions of: | |
|---|---|---|---|---|
| | | | **Intact fruits** | **Sliced fruit only** |
| Shared compounds | 3681-71-8 | Cis-3-hexenyl acetate | Indian almond, strawberry guava, peach, plum | Bugweed, coffee, loquat, mango, melon, star fruit, tomato |
| | 141-78-6 | Ethyl acetate | Bugweed, coffee, Indian almond, loquat, mango, peach, strawberry guava, tree tomato | Chilli, guava, melon, rose apple, star fruit, tomato |
| | 105-54-4 | Ethyl butyrate | Strawberry guava, rose apple, mango | Bugweed, melon, peach, plum, star fruit, tree tomato |
| | 105-37-3 | Ethyl propionate | Bugweed, rose apple, tomato, papaya | Guava, melon, plum, star fruit |
| | 142-92-7 | Hexyl acetate | Chilli, eggplant, Indian almond, papaya, peach, pumpkin, star fruit, strawberry guava | Plum |
| Species-specific compounds | 109-21-7 | Butyl butyrate | Plum | – |
| | 868-57-5 | Methyl 2-methylbutyrate | Rose apple | Melon, star fruit |
| | 106-73-0 | Methyl heptanoate | Star fruit | – |
| | 111-11-5 | Methyl octanoate | Peach, star fruit | Guava, strawberry guava |
| | 1191-16-8 | Prenyl acetate | Indian almond | Guava, strawberry guava |

The last column shows a list of fruit species in which the compound was found in sliced fruit samples only and not in intact fruit.

environment, which may include volatile emissions from non-host plants[3,4]. Their olfactory systems may be tuned to host-specific compounds. Some compounds shared between fruit species may also be emitted by other plant parts and non-host species. Therefore they may be widely distributed in the natural environment where the species evolved and, thus, be neglected by the insect. Cis-3-hexenyl acetate is a case in point. Although emitted by many fruit species, this compound is also emitted by leaves in response to herbivore damage[38]. It may have little ecological relevance for fruit feeding insects if any. In this scenario a strong antennal response would only be expected for shared compounds that are specifically emitted by fruits preferred by the insect. Using our dataset to test this would require classifying the sampled fruits according to the degree of attractiveness of their volatile emissions. Screening fruit attractiveness would therefore require a dedicated study. Still the hypothesis would have predicted a low response to shared plant compounds and species-specific host compounds rather than the negative correlation that we observed. Moreover shared compounds that are specific to host fruit should be attractive to insects, irrespective of the dose. We observed a preference to species-specific fruit compounds at low doses, which suggests that another factor may be involved.

Our results led us to consider the alternative hypothesis, namely that polyphagous insects have an ecological need to discriminate between host species. The ability to change host preference to adapt to a dynamic environmental context provides evolutionary advantages, which have been widely discussed[14,39,40]. Host preference in *Tephritidae* can change as demonstrated by the comparative analysis of wild and laboratory-reared individuals of the medfly species, *Ceratitis capitata*[41]. A switch in host preference can be induced in *C. capitata* females[13] and adult attraction behaviour can be conditioned[42] or influenced by larval diet[22] in *B. dorsalis*. As an adaptive consequence of learning host preference of *C. capitata* changes depending on host abundance[23], as predicted by Jaenike's optimal oviposition behaviour theory[39]. Optimising host use for reproductive fitness in the insect's environment should likely involves the ability to discriminate hosts, at some level. We therefore hypothesised that olfactory systems adapted to find and discriminate hosts that matter most for reproductive fitness were selected. We challenged the hypothesis that *Tephritidae* rely on species-specific fruit compounds to discriminate fruit species using a computational model of olfactory detection. The model suggested that ecological pressure to discriminate between fruit species would result in lower sensitivity thresholds for species-specific compounds compared to shared compounds. This phenomenon would generate a negative correlation between antennal response and the degree of sharedness among fruits.

Thus the accumulated experimental evidence suggests that olfactory systems have been shaped by two ecological needs: to detect a diversity of

hosts and to discriminate among them. Our computational model offers a plausible scenario to reconcile these needs: a higher proportion of shared compounds than species-specific compounds are detected, but with lower olfactory sensitivity. Our observations add an additional level of complexity to the olfactory strategies of polyphagous species. Polyphagous tephritids are clearly attracted to shared fruit compounds[8,9] and our behavioural assay further confirmed that they prefer these compounds over unshared ones, at least at a given dose. The low antennal response to shared compounds does not indicate that insects neglect them, but rather that high sensitivity is unnecessary since these compounds are frequently encountered in the insects' natural environment. At the same time polyphagous tephritids exhibit higher sensitivity to species-specific compounds, supporting the idea that they use these for host discrimination. Consistent with this we predicted a dose-dependent switch in preference, which is exactly what we observed. Comparing the dose at which the switch occurred to natural fruit emissions remains challenging since it consisted in a limited source of compounds that was slowly depleted by passive diffusion into still air, establishing a concentration gradient. In addition it is important to note that blends of only species-specific fruit compounds (i.e. with no shared compounds) do not exist in nature. Hence the observed preference shift can be considered a sensory illusion, which stems from an ability to discriminate between hosts. Further studies using more ecologically relevant fruit odours would be necessary to determine whether tephritids dynamically adjust their preference based on compound concentration under natural condition. To conclude we postulate that insects detect hosts through host-shared compounds and choose between hosts through species-specific compounds. More generally we propose that the three key functions, namely, host detection, host vs non-host selection and discrimination between hosts help shape the chemical tuning of insect olfactory systems.

The negative correlation we found between antennal response and the degree of sharedness among fruits concerned all the species we tested, although it appeared more prominent in the five polyphagous species than in the three oligophagous species. This suggests that this adaptation to the species' ecology was based, in part, on a conserved trait in *Tephritidae*. This echoes the reported attraction of *Z. cucurbitae* and *B. dorsalis* to a blend of compounds shared by different fruit species[8]. Two hypotheses can explain this. First the oligophagous species that we studied may not be truly oligophagous. The Cucurbitaceae specialist *Z. cucurbitae* is sometimes considered to be polyphagous[16], following reports of infestations in several fruit families including, Anacardiaceae, Caricaceae, Combretaceae, Moraceae, Solanaceae, etc.[24,43–46]. Infestations of other fruit families were also reported for Cucurbitaceae specialists from the Dacus genus[44,46] and for the Solanaceae specialist, *N. cyanescens*[24]. Secondly phylogenetic analysis of host range

diversity within the Dacini tribe revealed multiple episodes of host range switch, which involved broadening or reducing host range[16,18,47]. Since peripheral olfactory systems are generally conserved across *Tephritidae*[48], oligophagous species might have a polyphagous-type olfactory system due to phylogenetic inertia. To conclude if the olfactory strategies we observed are related to polyphagy, similar analyses should be conducted on truly monophagous species or on polyphagous frugivores from other insect orders.

Understanding the olfactory strategies adopted by phytophagous insects that allow them to recognise their host plant not only provides fundamental insights into ecology and evolution, but it will also facilitate the effective design of attractants for pest control[49]. When it comes to efficient trapping our data suggest that tephritids may not have a high sensitivity to a blend of compounds shared by different fruit species. Inversely a blend of species-specific fruit compounds may not be sufficiently attractive compared to a mixed blend of compounds emitted naturally by various host fruit species. Furthermore synergistic interactions among volatiles might also increase the attractiveness, for example between shared and unshared volatiles, or among species-specific components. Thus it would be interesting to take our study further, for example, by combining compounds shared by different fruit species with species-specific fruit compounds chosen from the most attractive fruit. This type of blend would be attractive at both high and low doses and, by extension, at any distance from the lure. Identifying the optimum ratio of compounds in the blend is also a challenge, given the variation between fruit emissions. The ratio of shared compounds in a blend could either imitate an attractive fruit or correspond to the ratio of compounds most shared by host fruit species.

In conclusion this study considers two non-exclusive hypotheses regarding the olfactory strategies developed by fruit flies, which allow them to recognise their plant hosts. Our study supports the view that tephritids' olfactory systems are not only designed to detect hosts, but also to discriminate between them, which means they are efficient and flexible in their search for hosts.

## Methods
### Insects
Female individuals of eight species of *Tephritidae* were used for the electrophysiological experiments: *Bactrocera dorsalis* (oriental fruit fly), *Bactrocera zonata* (peach fruit fly), *Ceratitis capitata* (medfly), *Ceratitis catoirii* (Mascarene fruit fly), *Ceratitis quilicii* (Cape fruit fly), *Dacus demmerezi* (Indian Ocean cucumber fruit fly), *Neoceratitis cyanescens* (tomato fruit fly) and *Zeugodacus cucurbitae* (melon fly). They originated from laboratory strains, which were reared according to previously published procedures[24,50]. The individuals selected for electrophysiology had lived for respectively 12–15, 28–32, 10–14, 21–26, 13–17, 11–15, 10–16 and 25–31 days after emergence and belonged to the following generations: 62–64th, 179–180th, 107–108th, 311–312th, 77–79th, 59th, 64–65th and 7th.

The behavioural experiments were conducted on female individuals of *B. dorsalis* from the 76 to 78th generations, 18–25 days after emergence.

### Plant materials and collection of volatile compounds
Fruit volatile compounds were collected using dynamic headspace extraction. We put individual or clusters of fruit into Nalophan sample bags (SenseTrading, Gröningen, Netherlands) that were 47 cm wide. The bags were sealed with plastic ties at both ends to create a headspace. The air content of the headspace was pulled through an adsorption tube filled with Tenax GR (Supelco, Bellefonte, USA) using a portable membrane pump (Spectrex PAS-500; Spectrex, Redwood City, CA, USA) set at 100 mL min$^{-1}$, for the 2 h immediately following sealing. No gas was injected in the bag. For each sample a negative control sample was performed with an empty Nalophan bag.

In this study we selected 28 fruit species considered to be potential hosts of *Tephritidae*[19] (Table 1). Headspace samples were extracted from intact fruit on the tree or from sliced fruit at a ripening stage suitable for oviposition. For each fruit species sliced or unsliced, 3–8 (median 7) replicates

were performed, with a total of 193 samples for intact fruit and 176 for sliced fruit.

### Gas chromatography–mass spectrometry
Tenax traps were stored at room temperature and analysed within 7 days using an automated thermal desorber (ATD, Turbomatrix 350, Perkin Elmer, USA), coupled with GC–MS (Clarus 580 GC and SQ8T MS, Perkin-Elmer, USA). The parameters of the ATD were as follows: primary desorption 5 min at 250 °C, flow 50 mL min$^{-1}$; no inlet split; up to an air monitoring trap kept at 2 °C; secondary desorption 1 mL min$^{-1}$ for 5 min and a constant flow of 3 mL min$^{-1}$. An Elite-5MS capillary column (60 m × 0.25 mm i.d., 0.25 μm film thickness, Perkin-Elmer, USA) was used for chromatography. The oven was programmed from 50 to 270 °C at 8 °C min$^{-1}$. Helium was used as a carrier gas with a constant flow of 0.21 mL min$^{-1}$. Masses m/z from 27 to 350 were scanned during the 30 min-long run, without solvent delay, at a frequency of 2 Hz (0.4 s scans + 0.1 s interscan).

### Triple electroantennography EAG$_3$ bioassays
Electrophysiology experiments were conducted using the EAG$_3$ technique[25] (Fig. 4A). EAG$_3$ improves the estimation of the insect's olfactory responses by simultaneously recording at three antennal positions. A live fly was immobilised in a 1000 μL pipette tip with its head exposed. The base of the head was fixed with dental wax (Enta Periphery Wax, Netherlands). For species of the Dacini tribe with elongated antennae the first two antennal segments, which do not have olfactory functions, were stabilised with a silicone adhesive elastomer 'Kwik-Sil' (World Precision Instruments, Inc., Sarasota, USA). Four electrodes consisting of a chloride silver wire inserted into a glass micropipette (tip diameter 1–2 μm) filled with an electrolyte solution (120 mM NaCl, 5 mM KCl, 1 mM CaCl$_2$, 4 mM MgCl$_2$ and 10 mM HEPES) were used to perform the EAG$_3$ recording. The recording electrodes were placed in the latero-proximal, medio-central and latero-distal regions of the right funiculus. The reference electrode was inserted in the middle of the left eye. The recording electrodes were connected to a four-channel differential AC amplifier (model 1700, A-M system, Carlsborg, USA) which amplified (gain × 1000) and filtered the signals between 1 and 1000 Hz. Data acquisition was fitted by digitalising the signals at 500 Hz with an acquisition card (NI 9215, National Instruments, France), associated with Labview software (National Instruments, France). Data were analysed using a current source density model to infer the total antennal response from the EAG signal recorded at three positions[25,51].

Stimuli were delivered through a glass chamber (20 cm × 5 mm i.dFID.). The tube had a small hole 5 cm upstream its outlet into which the odour source was inserted. A continuous humidified and charcoal-filtered airflow (20 mL s$^{-1}$) across the glass chamber reached the insect, which was 1 cm from the tube outlet. The insect was stimulated using an odour cartridge consisting of a Pasteur pipette with a square of filter paper loaded with 1 μL of a solution of a synthetic compound diluted at 10$^{-4}$ v/v or 10$^{-2}$ v/v in mineral oil. The Pasteur pipette was inserted into the small hole in the glass chamber and exposed to a puff of air for 200 ms (3 mL s$^{-1}$), that was regulated with an electro-valve (LHDA- 1233215-H, Lee Company, France) and controlled by a digital module (NI 9472, National Instruments, Nanterre, France) and Labview software (National Instruments, France). The experiment consisted of passing 30 compounds (listed in Supplementary Table, with selection criteria detailed in the Result section) individually in a random order with a time interval of 1 min between each stimulation. Four negative control stimulations (mineral oil, the solvent used for dilution) were performed: one initially and then after the 10th, 20th and final compound stimulations. A positive control stimulation (1-octen-3-ol, dilution v/v: 10$^{-4}$ in mineral oil) was also performed initially.

### Coupling chopped gas chromatograph-triple electro-antennography detector (GC-EAD$_3$)
For GC-EAD$_3$ experiments EAG$_3$ antennal preparation was used as a detector in parallel with the flame ionisation detector (FID). A fused silica

Restek Rxi-5 ms column (30 m × 0.32 mm i.d., 0.25 μm film thickness, Restek, Lisses, France) was used and heated as follows: 50 °C for 1 min, then 6.5 °C min⁻¹ to 200 °C and 40 °C min⁻¹ to 250 °C, held for 5 min. The temperature of the FID was set to 250 °C. Helium N60 was used as the carrier gas. To improve the signal-to-noise ratio the GC system was equipped with a D-Swafer Dean's switch (PerkinElmer, Inc., USA) at the column output, as described previously[25]. The device allows for the GC signal to be chopped with the alternate emission of the effluent to both the EAG$_3$ and FID detectors[25,52]. The chop frequency was regulated to 1 Hz with Labview software (National Instruments, France) and a digital module (NI 9472, National Instruments, Nanterre, France). The signal was demodulated at the chopping frequency (1 Hz) before a similar analysis as for the EAG$_3$ recordings[25]. The sample was fed into the glass chamber using a 3-metre-long heated transfer line (Antelia, Dardilly, France). To deliver the stimulations to the insect a short length of column at the output of the transfer line was inserted into the small hole in the glass chamber used in the EAG$_3$ experiments.

We manually injected 1 μL of solution which consisted of hexane loaded with a blend of synthetic compounds diluted at 10 ng μL⁻¹ or 1 μg μL⁻¹ each, into the GC. In total 37 synthetic compounds (Supplementary Table, with selection criteria detailed in the Result section) were split into two blends, which were delivered to the same individual in a random order. One run lasted 30 min i.e. one experiment lasted 1 h. For assessing the functional stability of the EAD preparation, negative (mineral oil) and positive (hexenyl acetate and 1-octen-3-ol, dilution v/v: 10⁻² in mineral oil) control stimulations were successively performed before and after each run, following the same procedures as for the EAG$_3$ experiments.

## Computational model of olfactory detection

Our modelling approach was designed to formally validate the two following hypotheses. (1) The detection hypothesis: for a given number of olfactory receptors (ORs), the polyphagous species' ecological need to detect hosts selects stronger olfactory responses to shared volatiles compared to species-specific host volatiles. (2) The alternative discrimination hypothesis: the polyphagous species' use of volatiles to discriminate between hosts selects stronger olfactory response to species-specific compounds compared to shared host compounds. The modelling workflow was as follows: we designed a large number of simple neuronal models of insect olfactory systems, each one responded to a random selection of compounds with random sensitivity. This set of models is intended to include all possible combinations that may have been reached through random mutations although only some are realistic under natural conditions. We then identified the subset of models under hypothetical selection and analysed their specific properties. To do this we fed samples of intact or sliced fruit volatile emissions into the model (see collection method above). By drawing on these data sets, we estimated how effectively each model detected or discriminated between fruit species.

More specifically each model included three categories of virtual compartments: compounds ($n = 665$ or 511, from sliced and intact fruit, respectively), ORs (one OR unit in the model represents the combined activity of all olfactory sensory neurons (OSNs) that express the same olfactory receptor) and total antennal activity (equivalent to EAG$_3$). An odour (input) was defined by a combination of compounds at different doses (in log units). Each OR was activated by a fixed number of compounds, whose identity was randomly selected from the full set of volatile compounds using random sampling without replacement (R function sample). Thus while some compounds did not activate any OR, others activated several ORs, depending on the random assignment. A compound is considered to have been detected if it activates at least one OR. Each compound dose/OR response relationship was defined by two parameters: a sensitivity threshold (drawn from a uniform distribution covering 5 log units) and a dynamic range width (drawn from a log-normal distribution of mean 3 and standard deviation 1). Since the input data were normalised by geometric mean, the unit itself is arbitrary. The range of 5 log units (0 –5) was chosen because it encompasses all compounds initially detected in fruit

emissions (see data analysis section). The activation $x_{ij}$ of an OR$_i$ by compound $j$ was 0 below threshold, 1 above threshold + dynamic range width and increased linearly between the two, following the function $x = $ (dose −threshold)/dynamic range width. The activity of individual OR units (OR$_i$) was calculated by combining all compound activations with the following equation: $1 - \Pi_j(1 - x_{ij})$, which ensured that the activity level remains between 0 and 1. This constraint reflects a well-established biological saturation effect, where OR units cannot be more than maximally activated (i.e. an activity of 1), corresponding to a plateau rather than a continuous increase in activity. The antennal activity (EAG) was the sum of OR activities, was not constrained in the model and varied depending on the compounds detected and their corresponding activation levels. The fact that each OR contribute homogeneously to antennal activity is a simplification unlikely to impact the amount of information available to the brain, since the input gain of each OSN expressing the same OR can be regulated independently in the antennal lobe's glomerulus.

For each model a fruit detectability index, which estimates the effectiveness of the insect sensory system to detect fruit, was defined as the sum of total antennal response to all fruit odours. The primary purpose of the index, which could be interpreted as a 'signal-to-noise ratio', is to provide a simple and intuitive estimation of the likelihood of the presence of fruit, regardless of the ORs involved in its detection. It can also be considered that a strong response ensures a response still occurs at a lower concentration. In order to estimate the ease with which the sensory system discriminates fruit species we calculated a fruit species discriminability index as follows. A random forest analysis was applied to classify the patterns of OR responses to volatile samples into the different fruit species. The model was implemented using the randomForest function in R (R package randomForest) with ORN response patterns as predictor variables and fruit species labels as the response variable. We set the number of tree to 1000 to ensure robust classification performance. The index was defined as the opposite of the out-of-bag scores i.e. the number of correctly classified samples divided by the total number of samples (Fig. 4B) and served as a proxy of the brain's ability to discriminate fruit on the basis of OR responses. The random forest analysis is not part of the neuronal simulation per se. It's a tool used to analyse the output of the neuronal models (i.e. the OR responses) and captures the amount of sensory information available for central processing in the brain. We also calculated a joint functionality index defined as the product of the fruit detectability and the fruit species discriminability indices, each centred on 0.5 and with standard deviation adjusted for values between 0 and 1.

We then built 12 sets of 10,000 models. Each set consisted of model that shared the same number of ORs (10 or 20, a rough estimate of the number of ORs likely to be sensitive to fruit odours in *Tephritidae* species) and a fixed number of compounds detected per OR (low, medium or high value, i.e. 10, 20 or 60 compounds per OR for models with 10 ORs, 5, 10 or 30 compounds per OR for models with 20 ORs, respectively). All models within a given set had the same OR number and the same parameter structure. However within each set, the parameters (e.g. sensitivity thresholds, dynamic range width and compounds activated by each OR) were drawn randomly for each model, ensuring variability in the parameter values across models. Volatile data for sliced or intact fruit was fed into the models.

## Behavioural assay setup

An ex situ trapping system was used to study the behavioural response of *B. dorsalis* to different blends of compounds. The system consisted of two handmade traps, which were placed in a rearing cage measuring 30 cm × 30 cm × 30 cm. The traps were made using a 30 mL wide mouth glass vial sealed with parafilm and lined with yellow coloured tape to guide the flies towards the traps. The lid was then perforated and a sectioned transparent 1000 μL pipette tip was inserted into the hole, serving as a conduit for the flies. A piece of filter paper loaded with 1 μL of the tested blend was placed at the bottom of the trap and the odour diffused passively. The two traps in the cage were loaded either with different blends or with a negative control (mineral oil only). Six cages were tested simultaneously with the same

combination of traps in different positions. In each cage 30 starved gravid female flies were released simultaneously and the number of flies in each trap was noted every 30 or 60 min. Each trap was used only once and the cages were thoroughly cleaned with soap between experiments.

For behavioural assays two blends of 5 compounds were tested. The first blend consisted of methyl 2-methylbutyrate, butyl butyrate, methyl heptanoate, methyl octanoate and prenyl acetate. The second blend consisted of ethyl acetate, cis-3-hexenyl acetate, hexyl acetate, ethyl propionate and ethyl butyrate. Lastly we used a blend combining all ten compounds. Preliminary work for calibrating compound ratios was carried out in the SLU laboratory at the Swedish University of Agricultural Sciences in order to homogenise the volatile emission rates in the blend before the behavioural assays. The blends were diluted in mineral oil and tested at two different doses (dilution v/v $10^{-4}$ and $10^{-3}$ of total compounds). The percentage of captured flies was analysed using generalised linear mixed-effects models (GLMM) with binomial error distribution (*glmer* function, R package *lme4*). Blends and doses were considered as fixed effects and the experimental date was considered a random effect.

## Data analysis

A pre-processing of GC–MS data was carried out with MZmine2 software[53]. Compound peaks were detected by ADAP algorithm[54] (S/N threshold 3, Wavelet Coeff. 3, minimum feature height 1,000,000, coefficient/area threshold 300, peak duration 0 to 0.2, RT wavelet 0 to 0.05 min), deconvoluted through blind source separation (window width 0.1, minimum number peaks 2, RT tolerance 0.025), RANSAC aligned (m/z tolerance 0.3, RT tolerance 0.1 min, automatic iteration no., minimum points no. 50%, threshold 0.1 min) and gap filled (m/z tolerance 0.3, RT tolerance 0.02 min). Each compound was tentatively identified according to its molecular weight and retention index (RI, calculated from the retention times of C6-C30 n-alkanes) and by comparing its mass spectra with NIST 14 library databases. The identities of compounds used in electrophysiological and behavioural assays were confirmed by co-injection with authentic standards in the GC–MS (Supplementary Table). Whenever a tentatively identified compound was found in the EPI suite software[55] we used it to predict its OH rate constant and estimated accordingly its AL, as previously described[26]. Alternatively for 27 out of the 40 synthetic compounds used in this manuscript, the OH rate constant was extracted from the NIST chemical kinetics database[56].

Further statistical analyses were conducted using R software[57]. When the quantity of compounds in the control samples was greater than in the samples with fruit, they were excluded from the analysis ($p < 0.05$, Wilcoxon test, *wilcox.test* function, R package *Stats*). Volatilomics data were then normalised by dividing each compound peak by the geometric mean of each corresponding sample then Box–Cox transformed (*boxcox* function, R package *MASS*, lambda = 0.014) and Pareto scaled (*paretoscale* function, R package *RFmarkerDetector*[58,59]).

For each compound we designed specific indices to quantify its degree of sharedness among fruit emissions. First we calculated the proportion of samples per fruit species in which the compound was detected. While this proportion was often 0 or 1, it occasionally took intermediate values when a compound was present in only a subset of samples. We then defined the sharedness indices as the Shannon diversity of the fruit species emitting the compounds applying the *diversity* function from the R package *vegan*[60] to these proportions. Notably for distributions where the proportions are 0 or 1, the index corresponds exactly to the logarithm of the number of fruits emitting the compound. For this calculation we used the volatile samples from intact (sharedness$_{IF}$[28]) and sliced (sharedness$_{SF}$[28]) fruit of 28 species, as well as from intact (sharedness$_{IF}$[13]) and sliced (sharedness$_{SF}$[13]) fruit from a subset of 13 species. The subset included species with a *B. dorsalis* infestation rate of above 5%, which were collected in Réunion during a recent campaign to monitor *Tephritidae* infestation (Table 1)[19]. The indices sharedness$_{IF}$[28] and sharedness$_{SF}$[28] can vary between a minimal value of 0 (if a single fruit species emits the compound) and a maximal value of 3.33 (when all 28 fruit species emit the compound). Similarly the indices sharedness$_{IF}$[13] and

sharedness$_{SF}$[13] can vary between a minimal value of 0 (when a single fruit species emits the compound) and a maximal value of 2.57 (when all 13 fruit species emit the compound). To measure the phylogenetic signal Abouheif's $C_{mean}$ index (*abouheif.moran* function, R package *adephylo*) was computed for each compound based on an autocorrelation approach[61,62]. To calculate the index the phylogeny of the 28 fruit species was generated by Bayesian phylogenetic inference using Markov Chain Monte Carlo (MCMC) methods[63]. The matrix of phylogenetic proximities was performed with the 'oriAbouheif' method (*proxTips* function, R package *adephylo*)[62]. It was calculated for the 28 species using volatile samples of intact ($C_{mean,IF}$[28]) and sliced ($C_{mean,SF}$[28]) fruit.

After a preliminary demodulation of the EAD$_3$ signal at the chopping frequency (1 Hz), we calculated the total antennal activity from EAG$_3$ and EAD$_3$ data using the current source density (CSD) model[25,51]. To apply the linear models antennal activity data were normalised by adding a constant value before applying a Box–Cox power transformation. The constant value was chosen to ensure that the Box–Cox transformation was log-equivalent. The normality of the resulting distribution was assessed using a Shapiro–Wilk test. For linear models individual identity was used as an independent variable. Bootstrap (1000 samples) was used to calculate 95% confidence intervals (reported in brackets in the Result section) and to test whether Pearson's correlation coefficients ($r$) or Pearson's partial correlation coefficients ($\rho$) differed significantly from 0 in a two-tailed test. Partial correlations were used to control for potential confounding variables and were computed as follows: First the effect of qualitative variables (e.g. individuals, chemicals classes, species) was removed from all the quantitative variables by fitting a linear model and extracting the residuals. Then the partial coefficient was computed by controlling for quantitative variables (e.g. atmospheric lifetime, depletion rate) using the pcor function from the RVAideMemoire package in R. For testing the effect of interactions linear regression or anova were used.

## Reporting summary

Further information on research design is available in the Nature Portfolio Reporting Summary linked to this article.

## Data availability

Data have been deposited in a long term repository service: CIRAD Dataverse, V1 (https://doi.org/10.18167/DVN1/T3XUXW)[64].

## Code availability

The code used for modelling has been deposited in a long term repository service: CIRAD Dataverse, V1 (https://doi.org/10.18167/DVN1/T3XUXW, CIRAD Dataverse, V1)[64].

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

## Acknowledgements

The authors would like to thank Serge Glenac and Jim Payet for collecting and rearing the insects. The authors greatly acknowledge the Plant Protection Platform for their technical support (3P, IBISA). This research was conducted within the frameworks of the research platform 'Biocontrôle et épidémio-surveillance végétale en Ocean Indien' (https://www.dp-biocontrole-oi.org/) and the UMT BAT 'Biocontrôle en Agriculture Tropicale'. This work was co-funded by the European Union and the Région Réunion. L'Europe s'engage à La Réunion avec le FEDER (European Regional Development Fund ERDF 2024-1248-005756). It was also funded by the Centre de Coopération Internationale en Recherche Agronomique pour le Développement (CIRAD) and the French Ministry of Agriculture, as part of the research project GEMDOTIS (Ecophyto II 2018) and ATTRACTIS (Ecophyto II 2023).

## Author contributions

G.R., M.C.M., P.F.D. and V.J. designed the study. G.R. collected the EAG/EAD and behavioural data and analysed all the data. M.C.M. collected the GC–MS data. V.J. contributed to data analysis and did the modelling. T.D., S.L.H. contributed to designing the behavioural tests. P.F.D., B.R. and V.J. supervised the work. G.R., V.J. wrote the manuscript. All authors edited the manuscript.

## Competing interests

The authors declare no competing interests.

## Additional information

**Supplementary information** The online version contains Supplementary material available at https://doi.org/10.1038/s42003-026-09751-3.

