## [Transparent Peer Review File · Communications Biology]

Olfaction in fruit flies (Tephritidae) balances detection and discrimination of host fruits

Corresponding Author: Mr Vincent Jacob

Version 0:

Reviewer comments:

Reviewer #1

(Remarks to the Author)

In this paper the authors set out to explore the chemical interactions between fruit flies and their host fruit species. It is a well-designed study, however, there are a few methodological details that are missing (details in the end). My main request from the authors is to publish the raw data (.cdf files of the mass spectrometry) in the repository. This open-science approach will enable to re-analysis the data using different analysis workflow, as well as using updated version of NIST and additional libraries.

My main question is regarding the initial choice of odor concentration for both the EAG3 and the behavioral assay. In nature (open air), I assume the detection concentration are lower. In Figure 5 it could be nice to have also 10-5 dilution to validate the model.

It have been found that odor threshold in humans is correlated with atmospheric lifetime for a given chemical family [Williams J, Ringsdorf A. 2020]. The assumption is that when VOCs are released to the outside air, their initial concentration will diminish with time and distance from the source as a result of mixing with cleaner surrounding air, and by photo-chemical oxidation. It could be interesting to test whether there is a correlation between the atmospheric lifetime and alpha-diversity indices, for esters and terpenoids(Figure S1). You can find those indecies in NIST kinetic database [16], and the IUPAC kinetic database.

Methodical details that are missing:

- >> [374-380] Is the 2 hour headspace sampling was immediately after the fruit was put into the sample bag?
- >> [376-377] what is the carrier gas for dynamic headspace sampling?
- >> [386] "Tenax traps were stored at room temperature and analysed within seven days". Can you indicate the changes from day 1 to day 7? Did you run replicates in diffent days?
- >> There is a missing paragraphs in the method regarding the details of the mass-spectrometer.
- >> [379-380] "For each sample, a negative control sample was performed with an empty nalophan". The air surrounding the trees might contain important molecules which stems from the fruit. And indeed, [523-524] "When the quantity of compounds in the control samples was greater than in the samples with fruit, they were excluded from the analysis". How many sampled were excluded?
- >> [436] The sample was fed into the glass chamber using a 3 m long heated transfer line. M is METRE?
- >> Please add the α -diversity indices to Table S3.
- >> Figure3C. Please add the Pearson coefficient to each plot.

Reviewer #2

(Remarks to the Author)

I found this paper very interesting indeed. The authors challenge a long-standing assumption that the olfactory system of polyphagous insects should be tuned primarily to volatiles that are common to most of their hosts (shared volatiles) as opposed to volatiles that are more species-specific. This is not an easy hypothesis to test because of the extensive host ranges of these insects (and probably why it has never been challenged as such). This bold study addresses this through an integrative approach using tephritid fruit flies as a model, bringing together odour analysis of an impressive diversity of hosts, electrophysiological analysis (EAD) to confirm (and estimate strength of response to) shared and species-specific volatiles, behavioural experiments testing mixed blends, and complex neurological modelling based on ecological theory. Their findings reveal some very interesting deviations from theory that I believe are important, even ground-breaking, in

progressing our understanding of olfaction in polyphagous insects. The paper certainly will influence thinking in this field.

Methods: I am not a modelling expert, by any means, so cannot comment on the methods employed here, but the hypotheses appear rational and in line with theory. The methods of odour analysis, EAD, and behavioural experiments are sound.

I enjoyed the discussion and found it very readable and well thought out. I have a few comments the authors might like to consider:

Rather than selection pressure being for an olfactory system that discriminates all hosts, I wonder if it might be more in line with behavioural ecology theory that selection pressure favours an olfactory system that optimises host use for reproductive fitness in the insect's environment; and that this likely involves host discrimination (at some level). May sound a bit nuanced, but with 100s of hosts, all varying in abundance and seasonally, the olfactory system may be adapting predominantly to finding and discriminating the host odours that matter most for reproductive fitness (noting also that the preference-performance hypothesis is always a bit shaky when tested empirically). If some hosts are rare, or if differences in fitness effects among certain hosts are small, there might not be selective pressure to distinguish between them.

Line 290 example with cis-3-hexenyl acetate, "little ecological relevance for [fruit feeding] insects" presumably, as this volatile is a known attractant for quite a few polyphagous insects.

For many fruit feeding tephritids, a fruit only becomes a host when it is ripe. Could this play a role in selection for particular shared volatiles – those which indicate ripe fruits.

Amplitudes of EAD responses are central to the hypotheses presented and tested in this study. I think to support this the discussion requires some justification that empirical evidence shows that strength of EAD responses generally equates to strength of behavioural response. Also to consider that EAD response (detection) is not the same as attraction; volatiles could be deterrents (either outright or at higher concentrations).

Line 474: "For each model, a fruit detectability index, which estimates the effectiveness of the insect sensory system to detect fruit, was defined as the sum of total antennal response to all fruit odours." Could the authors provide justification for this?

Synergistic interactions among volatiles might also be important; for example between the shared and unshared volatiles to maximise the olfactory perception of "(ripe) fruit", or among the more species-specific components to reduce the total number of volatiles required to discriminate the maximum number of hosts (combinational/configural effects). Might this need a mention?

Could the authors provide some info on the extent to which the "low" and "high" concentrations of volatiles used in the behavioural experiments equate to volatile production by fruits.

Could the authors clarify the justification for analysing damaged fruit? Line 67 states "Damaged fruit in orchards may emit new and potentially attractive compounds to fruit flies."; but these are not insects that require damaged fruit or have known fitness advantages to infesting damaged fruit, they prefer ripe fruit.

Abstract/Intro: While I see this as a great paper and would like to see it published, I did feel the abstract and introduction still need some work. I don't feel either clearly and succinctly address what this paper is about. The abstract rather strings together (albeit valid) statements and summarizes methods without clearly laying out the central theory / hypothesis, how this is tested and how the results challenge and progress theory (i.e. it's a bit messy and lacks flow). This is needed, particularly with a paper like this that integrates different methods, addresses and challenges theory, and is of interest to a wide audience.

Similarly for the intro, which I felt also lacks flow and build within and across paragraphs. In paragraph 2, I see the point that learning relies on an ability to discriminate among host odour, but not sure if this should be the central justification / evidence for discrimination in olfaction; what about, for example, mention of how these flies are known to discriminate among their hosts (loads of examples), quite markedly.

In outlining the hypotheses (line 62 onwards) I felt it again it didn't quite capture the essence of this study. It begins with "The present study focuses on the evolutionary pressure on the peripheral olfactory system that led to the development of a polyphagous strategy in eight Tephritidae fruit fly species with wide and overlapping host ranges: *B. dorsalis* [etc.]" I would rather see this more simply put in a way that is clear to readers; e.g. in terms of understanding how the olfactory system of polyphagous insects is tuned to species-specific volatiles vs volatiles that are common to all hosts in order to optimise host finding and host selection. Then I think stepping through the more specific aims and summarising how this was achieved.

Hopefully, with some polish to the Abstract and Intro this paper will be suitable for publication and I commend the authors on this exciting (and extensive) piece of research.

Reviewer #3

(Remarks to the Author)

The study by Ramiaranjatovo et al. represents an impressive amount of work on plant volatiles and their perception by

Tephritidae flies. The authors characterised fruit odours of different ripe fruits, and studied perception of compounds that are shared by many fruit species or unique to single species. They hypothesise a trade-off between detecting compounds and discriminating between different fruit species.

Unfortunately, I see several major issues with the argumentation and the rationale of the study. Therefore, I am not convinced that the main claim of the study – that there is a trade-off between detection ability and discrimination ability – is justified.

Firstly, using alpha diversity indices to measure the 'sharedness' of a compound between different fruits does not make sense. Alpha diversity is the diversity within a sample, beta diversity is the diversity between different samples. Thus, beta diversity is the metric of interest here. To quantify 'sharedness' you might even use a simple metric like the number of species that express it, but alpha diversity is entirely misleading in this context. I could not find out from the text which data actually entered the quantification of alpha diversity, and how it was calculated. Lines 529-537 do not provide enough details (and it is unclear why Jost 2007 is cited here). In Fig. 3 (legend) the authors refer to "fruit alpha diversity". It is unclear if diversity values are calculated for the bouquet of a single fruit, or for single compounds. This needs to be defined very clearly. Petrén et al. 2024 Ecological Monographs may be helpful for that.

Secondly, a main point of the study seems to be the negative correlation between antennal response and fruit alpha diversity. Looking at the graphs, I am not so sure how robust these results are (Fig. 3). Some correlations (especially *B. dorsalis*, *C. capitata*, *C. catovirii*) may be based on high responses for samples with near-zero diversity; I wondered whether they are still significant if these are left out. This can be tested using bootstrapping. Also, it should be tested against a more appropriate measure of diversity.

Thirdly, the results shown in lines 215-233 are interesting: female *B. dorsalis* display a dose-dependent switch in preference. However, unfortunately, this does not contribute much to corroborating the hypothesis of shared vs. species-specific compounds, because only two sets of compounds were tested. I am aware of the large effort to test this, but a dose-dependent switch may be due to multiple causes if we are talking about preferences of two sets only. Maybe one of the compounds in one blend happens to be slightly toxic at higher doses, but not at lower doses (which would not be unusual in plant volatiles). Thus, this experiment alone does not allow generalisation and conclusions on the general hypothesis. In line 386, you state that traps were kept at room temperature and analysed within 7 days. Can you confirm somehow that chemical composition was not affected by that? Usually, plant volatiles are highly reactive, and should either be analysed right away or kept at -20°C (or lower).

Finally, I am not convinced by the model. Note that it is a simulation model, not a neuronal model. It is important here to provide more details about the algorithms that produced the output (and e.g. the language used). Also, I missed a justification for the model assumptions. Does it make sense to enter the total antennal activity, and thus keep it constrained? Is there evidence from previous studies that total antennal activity is somehow constrained or constant in fly chemoreception?

Minor comments.

Line 48 Miridae are predatory bugs. Not sure why you are referring to them here.

Line 525-527 how often was this the case?

Fig. 1E What does the height of the bars represent? Please clarify.

Lines 132-140 Please provide details on the statistical models you used.

Line 194-196 Please explain the statistical model you used, and provide details on the interaction you found.

Line 218 what is meant by the statement that "the ability... has an ecological role"? Also, this should be in the discussion, not in the results section.

Line 420 Can you provide details of the selection criteria, that is, why these 30 compounds were chosen?

Line 422 Please show the data for the positive and the negative control as well. Can you provide evidence (or references) that mineral oil is an appropriate negative control?

Line 520 Please make clear in the tables that identification relied on comparison with libraries only and is therefore tentative. Different compounds may well have similar mass spectra and retention indices.

Version 1:

Reviewer comments:

Reviewer #1

(Remarks to the Author)

The authors have satisfactorily addressed all of my previous comments and concerns. I find the revised manuscript significantly improved, I only wish them to rephrase lines 104-105 and 392, as the reactivity of OH is considered chemical properties (atmospheric lifetime is not a chemical property)

Furthermore, line 105-106, "we also investigated whether this was the case for Tephritidae"

In Williams, J. et al, Molecules with short atmospheric lifetimes tend to be more sensitively detected by the human nose. You did not investigate the correlation between atmospheric lifetime and sensitivity (detection) of the fly but to the host fruit sharedness indices.

Lastly, Figure S2, Can you re-check if the x-axis values are correct?

Reviewer #2

(Remarks to the Author)

I have read through the response by the authors to my previous comments and thank them for their detailed reply. The changes that have been made to the revised manuscript address all these comments sufficiently; as such I am happy with this version. Although I do recommend one last look through the edited text as I did notice a few minor grammatical glitches.

Reviewer #4

(Remarks to the Author)

I was asked to focus my comments on the issues provided by reviewers 2 & 3 and the authors' responses. Therefore, I focus first on their comments before providing a few additional queries and concerns.

Overall, I agree with reviewers 2 & 3's concerns. Reviewer 2 mentions problems with the prose but also states they find the paper worthy of publication after revision; I agree with both points. The paper needs a significant re-write for clarity and not just minor tweaks. It is very difficult to follow. Additionally, reviewer 3 takes issue with 5 points. The authors have satisfactorily responded to points 1, 2, and 4. However, 3 and 5 remain poorly addressed. See below:

- 1) Diversity measures, which are well addressed by the authors.
- 2) The robustness of the antennal responses to sharedness index results. I am satisfied by the authors' response here
- 3) The limitations of the behavioral data used – here I still take issue. This is a very weak part of the paper.
- 4) Stability of volatile samples – exceptionally well addressed by the authors - kudos!
- 5) Weakness in the modelling – here I still take issue. The model is fine, but it is far less convincing to me than other aspects of the paper.

Other issues I find:

1) Did the authors validate that the chemicals used in their EAG and behavioral tests were in fact correctly identified in their GCMS analysis via co-injection of standards? If not, that MUST be done. If they have mis-identified compounds in the GCMS analysis, then all their EAG results are void. If they did confirm chemical identities, they should mention this clearly in the paper and include which chemicals were identified.

On this point, most (all?) compounds were *tentatively* identified only via RI values and NIST database matching. This is mentioned in the Methods section for data analysis, but the discussion and results do not report tentative identification. Carry this wording "tentatively" throughout the paper, at every mention, unless identities were proven.

2) I also have some concerns about some of the data used to feed the EAG and modeling work. The authors collected headspace volatiles from 28 species of host fruit. If the goal is to understand how the flies locate and discriminate between fruits, why not use host and non-host fruit? If all the fruit studied are host fruit, then how does their volatile emission inform us about how the flies avoid non-hosts?

As mentioned above, I agree with the previous reviewer #2's assessment that the paper lacks flow and is difficult to follow, and that this is particularly problematic for a paper that is attempting to challenge a theory. The abstract stands out as quite confusing. Unfortunately, the lack of flow and structure in the paper makes it difficult to understand. I have spent quite a few hours poring over the paper, most readers will not do this. This will harm the potential impact of the paper when published. In reading the paper, I get the sense that it has been edited several times, but not substantially rewritten with improved narrative structure and cohesion. Therefore, older parts of the manuscript don't tie in neatly with the new improvements. A confusing patchwork results. On this topic: Although in the conclusion, the authors point out that the two hypotheses of the paper are non-exclusive, most of the paper frames the two central hypotheses in conflict with one another. It seems true that it is both valuable to locate and discern between host fruits. What evidence is there that there is a trade off in this paper? Just the model? Again, this seems like a problem caused by patchy editing.

I'm also not convinced by their argument that the model used is a neuronal model. This is not my area of expertise, so perhaps I am misunderstanding or overly cautious/conservative in my assessment here. In their response to reviewer 3 they stated that "Indeed, the model simulates the activity of olfactory sensory neurons (OSNs) in the antenna, which respond to specific odorants." Yet, how do we know that the neurons/receptors are specific to only one odorant? Receptors can be promiscuous, binding a variety of structurally related volatiles. How does this factor into their model? I'm not convinced by their arguments here.

The authors' response to reviewer 3's comments on the behavioral data is also not satisfying. It seems a rather random assortment of chemicals were added and tested at arbitrary levels. As reviewer 3 mentions, this is not sufficient. It might have been better to add a rare compound to a more sharedness-compound-rich fruit sample and test responses there. Some data has apparently been removed from this version of the manuscript, as I see mention of a 10^{-5} dilution. Okay to keep this, but it needs to be integrated more clearly and very cautiously interpreted. One compound in their blend could be totally swaying these results and we cannot therefore interpret the results with clarity.

The strongest and most convincing element of this paper was the comparison of sharedness of compounds to EAG response. I would encourage the authors to focus their readers on this data as to me, it provides the central pillar of their paper, and all other elements are supporting.

Other more minor issues:

"Mechanically damaged fruit" is used in the abstract and elsewhere. In scientific parlance, this phrase has a specific meaning, referring to fruit that have suffered from impacts/bruises due to harvesting, processing, etc. But the fruit in this study were not damaged via impact, they were sliced. I suspect that different volatiles are emitted when fruit is sliced than when bruised but skin remains intact. Therefore, I do not think the term "mechanically damaged" should be used in the paper. Sliced is a perfectly clear and accurate descriptor.

Usually, you do not include your results in the introduction (lines 90-97). The entire paragraph beginning on line 89 should be moved to the discussion.

Line 469: "Sucked out" is too casual here.

Paragraph beginning on line 534 describing protocol for GC_EAG is confusing to me. Why use mineral oil as the negative control here? Shouldn't it be hexane? Did you really inject pure mineral oil on your GC?

Line 361-362: While the complementary GC-EAD analysis does provide extra evidence that the results are valid and driven by the chemicals as intended, this is not a certainty. GC separation very frequently fails to separate structurally related

chemicals, yielding co-eluting peaks. Therefore, without an analysis to verify purity of the standards used, this is not a refutation of the possibility of chemical impurities skewing results. Therefore, language in this sentence must be edited, removing "refute".

Line 363: What are volatile compound emitters? This could refer to almost anything so it is too general to have meaning. I also notice compounds included in the data set that look to be background and are not found in nature – trioxane, dioxane. Unknown number 273 is classified incorrectly.

Line 391, "It is important to note that while our modelling formally validates a hypothesis that is consistent with our experimental observations, it does not constitute proof per se." This sentence bothers me. Modelling is a tool that can support or generate hypotheses, but it does not in itself constitute evidence or data that can formally validate a hypothesis. This is circular reasoning. I also chafe at the latter part of the sentence "it does not constitute proof", yes, certainly it does not. No reasonable scientist would find that one paper is sufficient to prove anything, even if it were the best paper ever written. I think the authors' intent was to say something more along the lines of "While we have evidence that supports our hypothesis, more research is needed to explore XYZ interactions, because of factors like ABC not included here."

Version 2:

Reviewer comments:

Reviewer #4

(Remarks to the Author)

Thank you for taking the time to revise the manuscript and respond to reviewer concerns. I find the revised version much improved and all my concerns well addressed. Congratulations on this impressive study!

Revised Manuscript Number: COMMSBIO-24-6596A

“Olfaction in Tephritidae: a balance between fruit detection and discrimination”

Point by point responses to reviewer’s comment:

The reviewer’s comments have been very helpful to improve our work and prepare the revised version. We have considered all issues addressed and please find the detailed point by point answers below, highlighted in italic blue.

We sincerely hope that the revised version will be suitable for acceptance in Communications biology.

Reviewer #1 :

In this paper the authors set out to explore the chemical interactions between fruit flies and their host fruit species.

It is a well-designed study, however, there are a few methodological details that are missing (details in the end). My main request from the authors is to publish the raw data (.cdf files of the mass spectrometry) in the repository. This open-science approach will enable to re-analysis the data using different analysis workflow, as well as using updated version of NIST and additional libraries.

The raw data were previously included in the repository in the Thermo PerkinElmer proprietary format (.raw), which can be read by the open source software Mzmine. We have since converted these files to the open format netcdf, which may be more appropriate for sharing with the scientific community.*

My main question is regarding the initial choice of odor concentration for both the EAG3 and the behavioral assay. In nature (open air), I assume the detection concentration are lower.

To ensure laboratory control conditions, many aspects of the stimulation we used differed from natural condition. Nevertheless, we believe that at least the lowest dose we used should correspond approximately to concentrations detected naturally.

Typically, a fruit emits each compound at a dose ranging from 0.1 to 100 ng/h, the sum of all compounds reaching 100 to 500 ng/h (see for instance Miano et al. 2024 Heliyon 10:9).

For a given dose loaded onto filter paper, the concentration of compounds in the air varies greatly depending on the delivery method and the compound used. Sweeping it with a constant flow of air, as in the dynamic headspace method typically used to measure compound concentration, or in typical olfactometers, hastens compound depletion. In doing so, we observed a depletion of β -ocimene in about 30-40 minutes, with an exponential trend (Jacob et al., 2021, J. Chemical Ecology, 47:153-166).

In our bioassay, the concentration is trickier to estimate. The air was still, there was no active transportation of compounds, so a concentration gradient was established by passive diffusion. The flies displayed the same level of responsiveness after 4 hours, suggesting that the compounds were not depleted after this time. The filter paper was loaded with 10 μL of 10^{-4} dilution of a mixture of compounds, which amounts to around 200 ng of each compound, thus the compound emission rate was less than 50 ng/h, if not much lower.

We have now discussed the point in the manuscript, also following reviewer #2 comment, emphasizing that further study with more naturalistic fruit odour would be necessary to confirm the occurrence of such a switch under natural condition.

For EAG, compounds were volatilised in a small air compartment where the loaded filter paper was positioned, and this headspace was briefly blown to the insect preparation. Under similar conditions, Anderson et al. (2012) estimated that a 2 second stimulation puff with a dose consisting in 1 μL at dilution 10^{-4} typically contains 0.1 to 10 ng of compound, with a huge disparity between molecules. For GC-EAD, half of the 10 ng injected in the GC was delivered directly to the antenna during a 10 second column effluent. In both cases (EAG and GC-EAD), measured responses to some compounds were close to nil at these doses, suggesting that the antennae were not overstimulated. In any case, we believe that these considerations do not hamper the strength of our conclusion. The selection pressure exerted at low concentrations should have visible consequences at higher doses – and this was also suggested by the model.

In Figure 5 it could be nice to have also 10^{-5} dilution to validate the model.

While we agree with the reviewer that testing a lower dose would be interesting, we also acknowledge Reviewer #3's point that the behavioural test itself has certain limitations. To address this, we have revised the manuscript to clarify the scope and interpretation of the test. Given these changes, we believe that testing a 10^{-5} dilution is no longer necessary to support our conclusions, which have been refined as follows:

- *The theory that permeates literature suggests that polyphagous tephritids exhibit a behavioural preference for shared fruit compounds. Our results confirm this at high doses: *B. dorsalis* prefers shared fruit compounds over non-shared ones.*
- *Interestingly, our electrophysiological data indicate a stronger EAG response to non-shared fruit compounds, suggesting higher olfactory sensitivity to these compounds. This initially appeared contradictory to the behavioural preference. Thus, we questioned if under some conditions, specifically a lower dose, species-specific compounds might be preferred to shared compounds. We found that it's the case, which helps reconcile this apparent paradox.*
- *We therefore conclude that a behavioural preference for shared compounds does not necessarily contradicts a higher olfactory sensitivity to non-shared compounds.*

Additionally, considering the substantial experimental effort already devoted to the chemical and neurophysiological analyses, we believe that further behavioural testing should be limited to what is strictly necessary. However, we would be happy to further discuss this point if the reviewer has additional concerns.

It has been found that odor threshold in humans is correlated with atmospheric lifetime for a given chemical family [Williams J, Ringsdorf A. 2020]. The assumption is that when VOCs are released to the outside air, their initial concentration will diminish with time and distance from the source as a result of mixing with cleaner surrounding air, and by photo-chemical oxidation. It could be interesting to test whether there is a correlation between the atmospheric lifetime and alpha-diversity indices, for esters and terpenoids (Figure S1). You can find those indices in NIST kinetic database [16], and the IUPAC kinetic database.

We thank the reviewer for highlighting this interesting study, which presents a compelling correlation between human odour thresholds and atmospheric lifetime. While the proposed assumption is intriguing, we believe it remains speculative. Such an effect might influence correlations between these variables for shorter ALs (around an hour?) but does not fully explain the linear trend still observed over longer ALs (a day). Alternative hypothesis could also be considered, such as physical and chemical constraints in VOC synthesis or detection, favouring molecules that are more readily oxidised.

Following the reviewer's suggestion, we examined the relationship between sharedness, chemical class, and AL. Our analysis revealed intriguing correlations: the statistical distribution of AL among fruit volatile emissions appears heterogeneous. More precisely, we collected $k(\text{OH})$ constant values and calculated AL for the subset of compounds used in insect olfactory experiments. Additionally, we employed the EPI system program to predict AL for a larger dataset, yielding estimates for 322 compounds from intact fruit volatile samples and 381 compounds from sliced fruit volatile samples.

We observed a correlation between sharedness among fruit emissions and AL, which varies by chemical class. Specifically, sharedness and AL were positively correlated for esters, alcohols, ketones and aldehydes. However, terpenoids consistently exhibited low AL, without correlation to sharedness. Further exploration of chemical class influence on other metrics revealed that the previously reported correlation between sharedness and phylogenetic signal was primarily driven by terpenoids.

Altogether, these observations provide a broader perspective on how fruit volatile compounds were shaped by selection, with differing evolutionary responses across chemical classes. Finally, we confirmed that the antennal response of tephritids remained negatively correlated with sharedness even when considering the chemical class (ester or terpenoid) and the atmospheric lifetime.

We have included these points in the Result section, with respective comments in the Introduction and Discussion section.

Methodical details that are missing:

>> [374-380] Is the 2 hour headspace sampling was immediately after the fruit was put into the sample bag?

Yes it was. We have now specified it.

>> [376-377] what is the carrier gas for dynamic headspace sampling?

Air. Actually, it was a pull-only headspace sampling. We now specify that no gas was injected in the nalophan bags.

>> [386] “Tenax traps were stored at room temperature and analysed within seven days”. Can you indicate the changes from day 1 to day 7? Did you run replicates in different days?

The replicates of a fruit species were typically stored during the same duration before analysis, except for a few fruits that we have used to test an eventual degradation of the chemical sampling. Variability within the same fruit species was low compared to variability across species, and we did not observe any clear correlation between storage duration and changes in chemical composition, except possibly for strawberry guava.

We have now explicitly stated in the manuscript that some degree of sample degradation may have occurred. We also included the new analysis as supplementary material.

>> There is a missing paragraph in the method regarding the details of the mass-spectrometer.

We added the mass specifications.

>> [379-380] “For each sample, a negative control sample was performed with an empty nalophan”. The air surrounding the trees might contain important molecules which stems from the fruit. And indeed, [523-524] “When the quantity of compounds in the control samples was greater than in the samples with fruit, they were excluded from the analysis”. How many sampled were excluded?

130 compounds were removed from intact fruit volatile samples, and 151 compounds were removed from sliced fruit volatile samples. We have now specified it.

>> [436] The sample was fed into the glass chamber using a 3 m long heated transfer line. M is METRE?

Yes. The transfer line is longer than in most equivalent setups, permitting, occasionally, an independent use of the EAG and GC components. We checked that the quality of the GC-EAD coupling was acceptable using a photoionization device. This was published as supplementary material in Ramiaranjatovo et al., 2023. We have replaced “m” by “metre” to avoid any possible confusion.

>> Please add the α -diversity indices to Table S3.

We have done it.

>> Figure3C. Please add the Pearson coefficient to each plot.

We have done it.

Reviewer #2 :

I found this paper very interesting indeed. The authors challenge a long-standing assumption that the olfactory system of polyphagous insects should be tuned primarily to volatiles that are common to most of their hosts (shared volatiles) as opposed to volatiles that are more species-specific. This is not an easy hypothesis to test because of the extensive host ranges of these insects (and probably why it has never been challenged as such). This bold study addresses this through an integrative approach using tephritid fruit flies as a model, bringing together odour analysis of an impressive diversity of hosts, electrophysiological analysis (EAD) to confirm (and estimate strength of response to) shared and species-specific volatiles, behavioural experiments testing mixed blends, and complex neurological modelling based on ecological theory. Their findings reveal some very interesting deviations from theory that I believe are important, even ground-breaking, in progressing our understanding of olfaction in polyphagous insects. The paper certainly will influence thinking in this field.

We thank the reviewer for his praises.

Methods: I am not a modelling expert, by any means, so cannot comment on the methods employed here, but the hypotheses appear rational and in line with theory. The methods of odour analysis, EAD, and behavioural experiments are sound.

I enjoyed the discussion and found it very readable and well thought out. I have a few comments the authors might like to consider:

Rather than selection pressure being for an olfactory system that discriminates all hosts, I wonder if it might be more in line with behavioural ecology theory that selection pressure favours an olfactory system that optimises host use for reproductive fitness in the insect's environment; and that this likely involves host discrimination (at some level). May sound a bit nuanced, but with 100s of hosts, all varying in abundance and seasonally, the olfactory system may be adapting predominantly to finding and discriminating the host odours that matter most for reproductive fitness (noting also that the preference-performance hypothesis is always a bit shaky when tested empirically). If some hosts are rare, or if differences in fitness effects among certain hosts are small, there might not be selective pressure to distinguish between them.

We agree with the reviewer that this distinction makes sense, rectifying our oversimplified initial hypothesis. We have amended the discussion in this way.

Line 290 example with cis-3-hexenyl acetate, "little ecological relevance for [fruit feeding] insects" presumably, as this volatile is a known attractant for quite a few polyphagous insects.

Thanks for the precision. We have specified it.

For many fruit feeding tephritids, a fruit only becomes a host when it is ripe. Could this play a role in selection for particular shared volatiles – those which indicate ripe fruits.

We have introduced the notion in our discussion. However, our dataset cannot challenge this hypothesis, since a single maturation stage was studied per fruit species – corresponding to a ripening stage suitable to oviposition.

Amplitudes of EAD responses are central to the hypotheses presented and tested in this study. I think to support this the discussion requires some justification that empirical evidence shows that strength of EAD responses generally equates to strength of behavioural response. Also to consider that EAD response (detection) is not the same as attraction; volatiles could be deterrents (either outright or at higher concentrations).

We have included this point in the discussion.

Line 474: “For each model, a fruit detectability index, which estimates the effectiveness of the insect sensory system to detect fruit, was defined as the sum of total antennal response to all fruit odours.” Could the authors provide justification for this?

The primary purpose of the index is to provide an estimation of the likelihood of the presence of fruit, regardless of the ORs involved in its detection. Rather than introducing a complex formulation, we opted for a simple and intuitive measure, which could be interpreted as a “signal-to-noise ratio”: summing the total antennal response to a fruit odour. The stronger the antennal response, the most likely the detection. It can also be considered that a strong response at a given concentration ensures a response still occurs at a lower concentration. This approach aligns with the general reasoning of our model, which aims to describe broad trends rather than uncover specific mechanisms.

We hope this explanation, that we have included in the manuscript, clarifies our rationale.

Synergistic interactions among volatiles might also be important; for example between the shared and unshared volatiles to maximise the olfactory perception of “(ripe) fruit”, or among the more species-specific components to reduce the total number of volatiles required to discriminate the maximum number of hosts (combinational/configural effects). Might this need a mention?

We have mentioned it in the discussion.

Could the authors provide some info on the extent to which the “low” and “high” concentrations of volatiles used in the behavioural experiments equate to volatile production by fruits.

A detailed reply to reviewer #1 also addresses this question. In short, even if we have some clues that the concentrations used are compatible with the range of concentrations emitted by fruit, it’s difficult to really conclude. The air was perfectly still during the behavioural experiment, and a gradient of concentration was established by passive diffusion. A measure typically uses a dynamic headspace, which would hasten the depletion of the compounds from the source, and disrupt the gradient of concentration.

We have discussed this point, and added that further study with more naturalistic fruit odour would be necessary to confirm the occurrence of such a switch under natural condition.

Could the authors clarify the justification for analysing damaged fruit? Line 67 states “Damaged fruit in orchards may emit new and potentially attractive compounds to fruit flies.”; but these

are not insects that require damaged fruit or have known fitness advantages to infesting damaged fruit, they prefer ripe fruit.

*Thank you for your comment. However, we respectfully disagree with this particular point. Based on our field observations, we have frequently noticed that tephritid eggs are often laid on pre-existing mechanical injuries on fruits. An ongoing experiment currently conducted on our lab is based on this principle: sentinel fruits, pre-perforated with a needle, were placed in orchards and collected after 24 hours. Eggs of wild *B. dorsalis* were predominantly collected on the perforation sites.*

We have included a justification for this in the manuscript.

Abstract/Intro: While I see this as a great paper and would like to see it published, I did feel the abstract and introduction still need some work. I don't feel either clearly and succinctly address what this paper is about. The abstract rather strings together (albeit valid) statements and summarizes methods without clearly laying out the central theory / hypothesis, how this is tested and how the results challenge and progress theory (i.e. it's a bit messy and lacks flow). This is needed, particularly with a paper like this that integrates different methods, addresses and challenges theory, and is of interest to a wide audience.

We have edited the abstract according to the reviewer's advices.

Similarly for the intro, which I felt also lacks flow and build within and across paragraphs. In paragraph 2, I see the point that learning relies on an ability to discriminate among host odour, but not sure if this should be the central justification / evidence for discrimination in olfaction; what about, for example, mention of how these flies are known to discriminate among their hosts (loads of examples), quite markedly.

Thank you for your feedback. We believe that host choice could eventually be performed without discrimination, based on shared compounds only. Preference would be given to fruits that release the greatest number of these shared compounds. This led us to introduce learning as a possible justification.

Upon further consideration, we also thought of another justification. The diversity in host preference observed between polyphagous species sharing the same host range would be unlikely if shared compounds alone were driving host choice.

We have revised this section to clarify this and included both justifications.

In outlining the hypotheses (line 62 onwards) I felt it again it didn't quite capture the essence of this study. It begins with "The present study focuses on the evolutionary pressure on the peripheral olfactory system that led to the development of a polyphagous strategy in eight Tephritidae fruit fly species with wide and overlapping host ranges: *B. dorsalis* [etc.]" I would rather see this more simply put in a way that is clear to readers; e.g. in terms of understanding how the olfactory system of polyphagous insects is tuned to species-specific volatiles vs volatiles that are common to all hosts in order to optimise host finding and host selection. Then

I think stepping through the more specific aims and summarising how this was achieved.

We have revised the introduction to improve its clarity and flow, ensuring a more coherent progression from the general concepts of host detection and polyphagous strategies to the specific aims of our study, as per your suggestion.

Hopefully, with some polish to the Abstract and Intro this paper will be suitable for publication and I commend the authors on this exciting (and extensive) piece of research.

Reviewer #3 :

The study by Ramiaranjatovo et al. represents an impressive amount of work on plant volatiles and their perception by Tephritidae flies. The authors characterised fruit odours of different ripe fruits, and studied perception of compounds that are shared by many fruit species or unique to single species. They hypothesise a trade-off between detecting compounds and discriminating between different fruit species.

Unfortunately, I see several major issues with the argumentation and the rationale of the study. Therefore, I am not convinced that the main claim of the study – that there is a trade-off between detection ability and discrimination ability – is justified.

We thank the reviewer for this valuable feedback. We appreciate the comments provided and have addressed each of the points raised below. We hope that our responses will clarify the rationale of the study and demonstrate the validity of our main claim—that there is a trade-off between detection and discrimination ability.

Firstly, using alpha diversity indices to measure the ‘sharedness’ of a compound between different fruits does not make sense. Alpha diversity is the diversity within a sample, beta diversity is the diversity between different samples. Thus, beta diversity is the metric of interest here. To quantify ‘sharedness’ you might even use a simple metric like the number of species that express it, but alpha diversity is entirely misleading in this context. I could not find out from the text which data actually entered the quantification of alpha diversity, and how it was calculated. Lines 529-537 do not provide enough details (and it is unclear why Jost 2007 is cited here). In Fig. 3 (legend) the authors refer to “fruit alpha diversity”. It is unclear if diversity values are calculated for the bouquet of a single fruit, or for single compounds. This needs to be defined very clearly. Petré et al. 2024 Ecological Monographs may be helpful for that.

We sincerely thank the reviewer for highlighting the need to clarify our metric, previously referred to as “alpha-diversity.” We acknowledge that our initial description may not have fully conveyed its rationale and calculation, and we appreciate the opportunity to provide further clarification.

We have read Petré et al. 2024 with great interest. Their approach offers valuable methods for estimating the chemodiversity of compounds emitted by a single source, irrespective of

compound identity. Such measures, which yield a diversity index per fruit, are well suited for questions related to whether polyphagous insects prefer fruit odours with a high degree of diversity.

However, our research question differs in a fundamental way: we aimed to understand the ecological driving force shaping the olfactory tuning of polyphagous flies to individual compounds. Specifically, we sought to quantify the degree of sharedness of each compound among different fruit species. The index we chose is not much different from a simple presence/absence count, expressed in log scale.

More precisely, we used a measure derived from the “diversity” function in the R package vegan, based on the algorithm published in Jost 2007. We have clarified our calculation method in the manuscript as follows: For each fruit species, we first determined the proportion of samples in which a given compound was detected. While this proportion was typically 0 or 1, some compounds were detected in only a subset of samples. We then applied the “diversity” function (Shannon) to obtain an index that reflects the degree of sharedness of a compound among different fruit species. Notably, for distributions where the proportions are 0 or 1, the index corresponds exactly to the logarithm of the number of fruits emitting the compound.

After careful consideration of the reviewer’s suggestions, we have not identified a more suitable index for addressing our research question. However, to prevent confusion with existing chemodiversity metrics, we have renamed it the 'sharedness index' in the revised manuscript. We hope this clarification improves the understanding of our approach.

Secondly, a main point of the study seems to be the negative correlation between antennal response and fruit alpha diversity. Looking at the graphs, I am not so sure how robust these results are (Fig. 3). Some correlations (especially *B. dorsalis*, *C. capitata*, *C. catovirii*) may be based on high responses for samples with near-zero diversity; I wondered whether they are still significant if these are left out. This can be tested using bootstrapping. Also, it should be tested against a more appropriate measure of diversity.

We appreciate the reviewer’s concern regarding the robustness of the negative correlation between antennal response and sharedness among fruit emissions. To address this, we followed the reviewer’s advice with performing a bootstrap test of the Pearson’s coefficient (and of partial Pearson’s coefficients, where applicable, such as when considering the contribution of species or chemical classes). The p-values from these bootstrap tests remained significant, with minimal changes compared with the previous tests, and have been included in the figure 3, figure S3 and figure S4.

*In addition, in response to reviewer #1’s comment on atmospheric lifetime, we incorporated this variable along with chemical class (ester or terpenoid) as control variables. This adjustment led to the correlation of *B. zonata* antennal response becoming significant, while the antennal responses of the three oligophagous species lost their significance, with the index sharednessFC28. For the index sharednessFI28, only two species became non-significant. We have decided to include these tests in the Result section, while the simpler correlation was included in the figures.*

While we acknowledge that the correlation is weak, this is consistent with the inherent complexity of olfactory system tuning, which are influenced by many factors beyond just sharedness. Additionally, we would like to emphasise that the sharedness index we used is a

simplified approximation of the parameter supposedly under selection. As pointed out by Reviewer #2, the contribution of each fruit species to the fitness of the flies varies. Finally, the negative correlation was observed in several species, with two different stimulation doses, and using alternative indices of sharedness –reinforcing the robustness of our finding.

Thirdly, the results shown in lines 215-233 are interesting: female *B. dorsalis* display a dose-dependent switch in preference. However, unfortunately, this does not contribute much to corroborating the hypothesis of shared vs. species-specific compounds, because only two sets of compounds were tested. I am aware of the large effort to test this, but a dose-dependent switch may be due to multiple causes if we are talking about preferences of two sets only. Maybe one of the compounds in one blend happens to be slightly toxic at higher doses, but not at lower doses (which would not be unusual in plant volatiles). Thus, this experiment alone does not allow generalisation and conclusions on the general hypothesis.

We fully agree that our behavioural assay alone does not prove the broader hypothesis. Our goal was to design a minimal behavioural test that could suggest a potential ecological implication of the olfactory tuning we observed, complementing the substantial experimental effort already devoted to chemical and neurophysiological analyses.

That said, we believe the experiment still provides meaningful insights, and we have reformulated the argumentation in the manuscript as follows:

- *Previous studies (e.g. Biasazin et al. 2019) have shown that polyphagous tephritids display a behavioural preference for shared fruit compounds. Our results confirm this at a given dose: *B. dorsalis* prefers shared fruit compounds over species-specific ones.*
- *Interestingly, our electrophysiological data indicate a stronger EAG response to species-specific fruit compounds, suggesting higher olfactory sensitivity to these compounds. This initially seemed contradictory to the behavioural preference.*
- *To resolve this, we formulated an a priori prediction: under certain conditions, specifically at lower doses, species-specific compounds might become more attractive than shared compounds [Note that this was not merely a post hoc rationale: we genuinely predicted this switch before conducting the experiments].*
- *When we tested this prediction experimentally, the results aligned with our expectation.*
- *This finding suggests that a behavioural preference for shared compounds does not necessarily contradicts a higher olfactory sensitivity to species-specific compounds.*

Finally, regarding the possibility of toxicity influencing the results: as far as we know, it is unlikely that a shared compound in the blend would become more toxic at a lower dose. Conversely, if a species-specific compound becomes repellent at higher dose, we still demonstrated that it was more attractive than the shared compounds at low dose, which, in itself, was an unexpected finding that provides deviation from the theoretical framework previously envisioned.

In line 386, you state that traps were kept at room temperature and analysed within 7 days. Can you confirm somehow that chemical composition was not affected by that? Usually, plant volatiles are highly reactive, and should either be analysed right away or kept at -20°C (or lower).

We fully agree with the reviewer's concern, and we recognized that storing samples at room temperature was not ideal. To assess the potential impact of this, we analysed the variability of fruit emissions for the few fruits where samples were analysed at two different time points. Variability within the same fruit species was low compared to variability across species, and we did not observe any clear correlation between storage duration and changes in chemical composition, except possibly for strawberry guava.

We have now explicitly stated in the manuscript that some degree of sample degradation may have occurred. We also included the new analysis as supplementary material.

Finally, I am not convinced by the model. Note that it is a simulation model, not a neuronal model. It is important here to provide more details about the algorithms that produced the output (and e.g. the language used). Also, I missed a justification for the model assumptions. Does it make sense to enter the total antennal activity, and thus keep it constrained? Is there evidence from previous studies that total antennal activity is somehow constrained or constant in fly chemoreception?

We appreciate the reviewer's comment and understand the concern regarding the terminology, the algorithms, and the constraints in the model.

Regarding the terminology, we initially questioned it ourselves before concluding that the model can be considered a "neuronal model". Indeed, the model simulates the activity of olfactory sensory neurons (OSNs) in the antenna, which respond to specific odorants. While the model does not include synaptic interactions—consistent with the nature of peripheral olfactory systems—each 'OR unit' in the model represents a neuronal response to a set of compounds with a typical dose-response curve. The random forest analysis is not part of the neuronal simulation per se; it's a tool used to analyse the output of the neuronal model (i.e., the OR responses) and quantify how well the sensory system can detect and discriminate between odours. This approach aims to capture the amount of sensory information available for central processing in the brain. We have clarified this point in the manuscript.

The model was written with R, and all the algorithms have been included in an open-source long term repository. We have provided more details about the algorithms in the main text, and believe it is sufficient to reproduce the algorithm in any programming language. If needed, we can provide additional clarifications to the implementation.

To clarify the level of constraint, the activity of individual OR units is constrained between 0 and 1 to reflect a biological saturation effect, where OR units cannot be more than maximally activated (i.e., an activity of 1). This reflects well-established phenomena in insect olfaction, where strong stimulations result in a plateau in response rather than a continuous increase in activity. However, the total antennal activity (EAG) is not constrained in the model. It is the sum of all OR responses, which varies depending on the compounds detected and their corresponding activation levels.

That said, the assumption in our model that all OR units have the same maximum activity (set to 1) is indeed a simplification, since the number of OSNs per OR is heterogeneous in reality. However, this simplification is unlikely to impact the amount of information available to the brain, since the input gain of each set of OSN expressing the same OR can be regulated independently in the antennal lobe's glomerulus.

We have revised the manuscript to make this distinction clearer and provide additional context regarding the use of this constraint for individual OR activities.

We sincerely appreciate the reviewer's constructive feedback and believe that these clarifications will strengthen the manuscript.

Minor comments.

Line 48 Miridae are predatory bugs. Not sure why you are referring to them here.

Miridae is a highly diversified family of insects. Some are predatory, other phytophagous, and others have mixed diet. The article we cited clearly refers to phytophagous Miridae. We have specified it.

Line 525-527 how often was this the case?

130 compounds were removed from intact fruit volatile samples, and 151 compounds were removed from sliced fruit volatile samples. We have now specified it.

Fig. 1E What does the height of the bars represent? Please clarify.

It is written in the figure legend: "Each colour represents the chemical classes of compounds and the corresponding height is proportional to the number of compounds in each chemical class."

Lines 132-140 Please provide details on the statistical models you used.

We have now provided more details on the statistical models used. Specifically, we employed linear regressions or ANOVA, depending on whether the variables were qualitative or quantitative. Additionally, we now use a two-tailed bootstrap test to assess that the correlation coefficient significantly differs from 0. However, linear regressions or ANOVA are still used to estimate the interaction effect between two variables.

Line 194-196 Please explain the statistical model you used, and provide details on the interaction you found.

We have done it. We also explained the interaction. On figure 4C-D, the interaction is reflected in the curvature of the white iso-level lines, which are bend rather than straight. This effect also underlies the structure of the joint distribution.

Line 218 what is meant by the statement that "the ability... has an ecological role"? Also, this should be in the discussion, not in the results section.

The original sentence was indeed clumsy, and we have replaced it with a more descriptive transition. The ecological role has been detailed in the discussion.

Line 420 Can you provide details of the selection criteria, that is, why these 30 compounds were chosen?

We have now specified that the selection criteria are detailed in the Result section.

Line 422 Please show the data for the positive and the negative control as well. Can you provide evidence (or references) that mineral oil is an appropriate negative control?

Mineral oil is a standard negative control for this experiment, as it is the solvent used for dilution. We have specified it. Stimulating with mineral oil is equivalent to an infinite dilution, ensuring that any observed responses are due to the odorant and not the solvent.

Upon further review, we realize that the negative and positive control stimulations were included before and after the GC-EAD experiment, but the protocol differed for the EAG experiment. For the EAG experiment, we performed four negative control stimulations (initially, after the 10th, 20th, and final compound stimulations) and a single positive control stimulation at the beginning of the experiments. We have corrected the methods section accordingly and included the appropriate data for both experiments as requested.

Line 520 Please make clear in the tables that identification relied on comparison with libraries only and is therefore tentative. Different compounds may well have similar mass spectra and retention indices.

Thanks for the precision which we now have included in the manuscript.

Reviewers' comments:

Reviewer #1 (Remarks to the Author):

The authors have satisfactorily addressed all of my previous comments and concerns. I find the revised manuscript significantly improved.

We thank the reviewer for this positive feedback.

I only wish them to rephrase lines 104-105 and 392, as the reactivity of OH is considered chemical properties (atmospheric lifetime is not a chemical property)

We agree with the reviewer, and we have corrected the wording accordingly.

Furthermore, line 105-106, "we also investigated whether this was the case for Tephritidae" In Williams, J. et al, Molecules with short atmospheric lifetimes tend to be more sensitively detected by the human nose. You did not investigate the correlation between atmospheric lifetime and sensitivity (detection) of the fly but to the host fruit sharedness indices.

We thank the reviewer for noticing this. We have revised the lines as follow:

"Since the atmospheric lifetime (AL) of volatile compounds, which depends on their chemical nature, has been shown to correlate with their ecological significance, we also investigated whether the degree of sharedness among fruit correlates with AL." Following reviewer #4 suggestion to reorganize the introduction, this line has been moved to the "Result" section, as we thought it might not be central to understanding the basic principles of the manuscript.

Lastly, Figure S2, Can you re-check if the x-axis values are correct?
It is correct. We replaced 10^0 by 1 day, for simplification.

Reviewer #2 (Remarks to the Author):

I have read through the response by the authors to my previous comments and thank them for their detailed reply. The changes that have been made to the revised manuscript address all these comments sufficiently; as such I am happy with this version. Although I do recommend one last look through the edited text as I did notice a few minor grammatical glitches.

We thank the reviewer for their positive evaluation of our revised manuscript. We carefully proofread the text once more and corrected the minor grammatical issues.

Reviewer #4 (Remarks to the Author):

I was asked to focus my comments on the issues provided by reviewers 2 & 3 and the authors' responses. Therefore, I focus first on their comments before providing a few additional queries and concerns.

Overall, I agree with reviewers 2 & 3's concerns. Reviewer 2 mentions problems with the prose but also states they find the paper worthy of publication after revision; I agree with both points. The paper needs a significant re-write for clarity and not just minor tweaks. It is very difficult to follow.

Additionally, reviewer 3 takes issue with 5 points. The authors have satisfactorily responded to points 1, 2, and 4. However, 3 and 5 remain poorly addressed. See below:

- 1) Diversity measures, which are well addressed by the authors.
- 2) The robustness of the antennal responses to sharedness index results. I am satisfied by the authors' response here
- 3) The limitations of the behavioral data used – here I still take issue. This is a very weak part of the paper.
- 4) Stability of volatile samples – exceptionally well addressed by the authors - kudos!
- 5) Weakness in the modelling – here I still take issue. The model is fine, but it is far less convincing to me than other aspects of the paper.

We thank the reviewer for their positive evaluation of our responses to points 1, 2 and 4. We have carefully considered the remaining concerns regarding points 3 and 5, and provide detailed responses below.

Other issues I find:

- 1) Did the authors validate that the chemicals used in their EAG and behavioral tests were in fact correctly identified in their GCMS analysis via co-injection of standards? If not, that MUST be done. If they have mis-identified compounds in the GCMS analysis, then all their EAG results are void. If they did confirm chemical identities, they should mention this clearly in the paper and include which chemicals were identified.

We thank the reviewer for this important comment. We have injected standards for all compounds used in the EAG/GC-EAD/behavioural assays into the GCMS. The RI of most compounds identified in the samples closely matched those of the standard (difference < 10 units), confirming their identities. However, the identity of three compounds remained uncertain. In the GC-EAD assay, the RI of ethyl methacrylate standard deviated substantially from that observed in the sample, and only one isomer of camphene could be confirmed. In the

case of β -caryophyllene used in GC-EAD and GC-MS, the RI matched the standard in sliced fruit samples but not in intact fruit samples (difference = 19 units). Thus, these compounds were removed from the corresponding analysis. These exclusions did not alter the statistical outcomes. We have now clarified this procedure in the revised manuscript., which further strengthened our confidence in the reliability of our findings.

On this point, most (all?) compounds were *tentatively* identified only via RI values and NIST database matching. This is mentioned in the Methods section for data analysis, but the discussion and results do not report tentative identification. Carry this wording “tentatively” throughout the paper, at every mention, unless identities were proven.

We have added the term “tentatively” in the Results and Discussion sections wherever it applies. Compounds whose identity were confirmed by co-injection with authentic standards in the GC-MS are no longer referred to as tentative.

2) I also have some concerns about some of the data used to feed the EAG and modeling work. The authors collected headspace volatiles from 28 species of host fruit. If the goal is to understand how the flies locate and discriminate between fruits, why not use host and non-host fruit? If all the fruit studied are host fruit, then how does their volatile emission inform us about how the flies avoid non-hosts?

*The reviewer raises a relevant point. In our dataset, we sorted the sampled fruits according to their “host” status for *B. dorsalis*. Thirteen fruits were classified as major hosts, and were used to compute the sharednessIF13 and sharednessSF13 indices. However, identifying true “non-hosts” is more challenging for several reasons. First, *B. dorsalis* was occasionally found to emerge from most of the fruits collected in an extensive field campaign (Moquet et al. 2021), making it difficult to establish strict non-host categories. Second, host status is influenced not only by olfactory cues but also by non-olfactory factors, such as fruit epidermis thickness, presence of hairs, or larval performance. The preference-performance relationship is quite poor in polyphagous Tephritidae (Charlery de la Masselière et al. 2017). Addressing how flies avoid non-host fruits would indeed be highly valuable, but it would require a separate study specifically designed to evaluate the attractiveness of volatiles across a broad panel of fruits. We have noted this point in the discussion.*

As mentioned above, I agree with the previous reviewer #2’s assessment that the paper lacks flow and is difficult to follow, and that this is particularly problematic for a paper that is attempting to challenge a theory. The abstract stands out as quite confusing. Unfortunately, the lack of flow and structure in the paper makes it difficult to understand. I have spent quite a few hours poring over the paper, most readers will not do this. This will harm the potential impact of the paper when published. In reading the paper, I get the sense that it has been edited several times, but not substantially rewritten with improved narrative structure and cohesion. Therefore, older parts of the manuscript don’t tie in neatly with the new improvements. A confusing patchwork results.

We have revised both the Abstract and the Introduction to enhance the narrative structure and cohesion. We would also like to point out that we adopted a format commonly used in Neurophysiology articles, in which the Introduction section includes a brief statement

summarizing the key results. We felt that this approach is particularly relevant for presenting the behavioural tests, which were specifically designed on the basis of the GC/EAG results.

On this topic: Although in the conclusion, the authors point out that the two hypotheses of the paper are non-exclusive, most of the paper frames the two central hypotheses in conflict with one another. It seems true that it is both valuable to locate and discern between host fruits. What evidence is there that there is a trade off in this paper? Just the model? Again, this seems like a problem caused by patchy editing.

The reviewer is correct that the trade-off was only inferred from the computational model, and should be considered a hypothesis at this stage. However, we believe this hypothesis provides a plausible framework to reconcile the experimental data from this study (i.e., stronger EAG response amplitude to species-specific compounds) with those of Biasazin et al. 2018 (i.e., higher detection probability of shared compounds). We hope that the revised narrative clarifies this point.

I'm also not convinced by their argument that the model used is a neuronal model. This is not my area of expertise, so perhaps I am misunderstanding or overly cautious/conservative in my assessment here.

We appreciate the reviewer's concern, and have carefully reconsidered the terminology used in the revised manuscript. Specifically, we now distinguish between the global "computational model", which encompasses 12,000 individual "neuronal models". We believe this terminology more accurately reflects the nature of our modelling approach. Should the reviewer remain concerned, we would be glad to further refine the terminology.

In their response to reviewer 3 they stated that "Indeed, the model simulates the activity of olfactory sensory neurons (OSNs) in the antenna, which respond to specific odorants." Yet, how do we know that the neurons/receptors are specific to only one odorant? Receptors can be promiscuous, binding a variety of structurally related volatiles. How does this factor into their model? I'm not convinced by their arguments here.

We apologize for the lack of clarity. In the model, neurons/receptors are not assumed to be specific to a single odorant, rather, they can respond to multiple odorants. This was intended to be conveyed by our use of the plural form in our earlier response to Reviewer 3, but the wording may have been unclear. We would like to emphasize, however, that this point is already stated clearly in the manuscript.

The authors' response to reviewer 3's comments on the behavioral data is also not satisfying. It seems a rather random assortment of chemicals were added and tested at arbitrary levels. As reviewer 3 mentions, this is not sufficient. It might have been better to add a rare compound to a more sharedness-compound-rich fruit sample and test responses there. Some data has apparently been removed from this version of the manuscript, as I see mention of a 10^{-5} dilution. Okay to keep this, but it needs to be integrated more clearly and very cautiously interpreted. One compound in their blend could be totally swaying these results and we cannot therefore interpret the results with clarity.

We appreciate the reviewer's insightful comments and agree that testing behaviour with ecologically realistic blends could provide valuable evidence and may further strengthen our conclusions.

Our approach was, however, rooted in Behavioural Neurosciences, and consisted in designing controlled artificial stimuli to probe the behavioural correlates of specific physiological mechanisms revealed by our electrophysiological analysis. Specifically, the behavioural tests were designed as a direct follow-up to our EAG findings, which led us to hypothesize that attraction to shared compounds alone may require higher concentrations than when species-specific compounds are involved.

We therefore deliberately constructed a blend composed only of species-specific compounds. Although such a blend is not ecological, this design allowed us to isolate the effect of these compounds and demonstrate that attraction cannot be explained by the presence of shared compounds. Consistently, the assay supports the idea that some species-specific and some shared compounds can be behaviourally relevant for the insect, albeit under different conditions, consistent with distinct selection pressure on these two categories.

We have clarified this rationale and its limitations in the revised Introduction/Results/Discussion sections. Also, we would like to clarify that no data were removed from this version of the manuscript. After carefully reviewing the text, we could not find any mention of a 10^{-5} dilution. This may be a misunderstanding; if not, we would be grateful if the reviewer could kindly indicate the specific passage they are referring to, so that we can double-check.

The strongest and most convincing element of this paper was the comparison of sharedness of compounds to EAG response. I would encourage the authors to focus their readers on this data as to me, it provides the central pillar of their paper, and all other elements are supporting.

We fully agree with the reviewer's assessment. In the revised manuscript, we have emphasized that the comparison of sharedness of compounds to EAG response represents the central finding of the study. We hope that this is now clearer to the reader.

Other more minor issues:

“Mechanically damaged fruit” is used in the abstract and elsewhere. In scientific parlance, this phrase has a specific meaning, referring to fruit that have suffered from impacts/bruises due to harvesting, processing, etc. But the fruit in this study were not damaged via impact, they were sliced. I suspect that different volatiles are emitted when fruit is sliced than when bruised but skin remains intact. Therefore, I do not think the term “mechanically damaged” should be used in the paper. Sliced is a perfectly clear and accurate descriptor.

We have used the term “sliced fruit” throughout the revised manuscript.

Usually, you do not include your results in the introduction (lines 90-97). The entire paragraph beginning on line 89 should be moved to the discussion.

As explained above, Neurophysiology articles typically introduce the key results at the end of the Introduction. Although our study sits at the interface of Neurophysiology and Ecology, we felt that this approach was appropriate, as it helps guide the reader through the uncommon

rationale of the study, particularly regarding the purpose of the behavioural assay. Still, we introduced our main result as an overview, without presenting detailed data.

Line 469: “Sucked out” is too casual here.

“Sucked out” has been replaced with the more formal “pulled through”.

Paragraph beginning on line 534 describing protocol for GC_EAG is confusing to me. Why use mineral oil as the negative control here? Shouldn't it be hexane? Did you really inject pure mineral oil on your GC?

The GC-EAD protocol we used is based on a recent improvement of the coupling, described in detail and validated by Myrick and Baker (2018) and Ramiaranjatovo et al. (2023). In this study, we summarize the main principles, and we refer the reader to these papers for a more detailed description.

Concerning the controls, hexane itself would not be an appropriate negative control for the EAD responses to compounds that had already been separated from their hexane solvent by the GC. The positive control stimulations performed before and after the run were intended solely to verify that the EAG preparation remains functional and stable. For these controls, we used a standard stimulation: small compounds diluted mineral oil, with mineral oil alone serving as the corresponding negative control. We have clarified this purpose in the Methods section.

Line 361-362: While the complementary GC-EAD analysis does provide extra evidence that the results are valid and driven by the chemicals as intended, this is not a certainty. GC separation very frequently fails to separate structurally related chemicals, yielding co-eluting peaks. Therefore, without an analysis to verify purity of the standards used, this is not a refutation of the possibility of chemical impurities skewing results. Therefore, language in this sentence must be edited, removing “refute”.

We agree that the term “refute” was too strong and have revised the sentence accordingly. It now reads: “Artefactual responses to impurities have been reported, but our GC–EAD analyses of the same compounds make this explanation unlikely.”

Line 363: What are volatile compound emitters? This could refer to almost anything so it is too general to have meaning.

We have replaced it with “sources of volatile compounds”, which is a more precise and academically appropriate term.

I also notice compounds included in the data set that look to be background and are not found in nature – trioxane, dioxane.

We thank the reviewer for this comment. Trioxane and dioxane were detected in our GC-MS analyses. While these compounds are not naturally occurring in our studied samples and may represent background or laboratory contaminants, we have chosen to retain all detected compounds in the dataset for full transparency. Our main analyses and conclusions focus on

naturally occurring compounds relevant to the study, but we report all observations to allow readers to fully evaluate the data.

Unknown number 273 is classified incorrectly.

We thank the reviewer for pointing this out. The compound has now been reclassified as “unknown”.

Line 391, “It is important to note that while our modelling formally validates a hypothesis that is consistent with our experimental observations, it does not constitute proof per se.” This sentence bothers me. Modelling is a tool that can support or generate hypotheses, but it does not in itself constitute evidence or data that can formally validate a hypothesis. This is circular reasoning. I also chafe at the latter part of the sentence “it does not constitute proof”, yes, certainly it does not. No reasonable scientist would find that one paper is sufficient to prove anything, even if it were the best paper ever written. I think the authors’ intent was to say something more along the lines of “While we have evidence that supports our hypothesis, more research is needed to explore XYZ interactions, because of factors like ABC not included here.”

We thank the reviewer for this comment. We have removed the sentence, which was included out of an excess of caution.